# Overexpression screen of interferon-stimulated genes identifies RARRES3 as a restrictor of *Toxoplasma gondii* infection

Nicholas Rinkenberger[1], Michael E Abrams[2], Sumit K Matta[1], John W Schoggins[2], Neal M Alto[2], L David Sibley[1]*

[1]Department of Molecular Microbiology, Washington University in St. Louis, St Louis, United States; [2]Department of Microbiology, University of Texas Southwestern, Dallas, United States

**Abstract** *Toxoplasma gondii* is an important human pathogen infecting an estimated one in three people worldwide. The cytokine interferon gamma (IFNγ) is induced during infection and is critical for restricting *T. gondii* growth in human cells. Growth restriction is presumed to be due to the induction of interferon-stimulated genes (ISGs) that are upregulated to protect the host from infection. Although there are hundreds of ISGs induced by IFNγ, their individual roles in restricting parasite growth in human cells remain somewhat elusive. To address this deficiency, we screened a library of 414 IFNγ induced ISGs to identify factors that impact *T. gondii* infection in human cells. In addition to IRF1, which likely acts through the induction of numerous downstream genes, we identified RARRES3 as a single factor that restricts *T. gondii* infection by inducing premature egress of the parasite in multiple human cell lines. Overall, while we successfully identified a novel IFNγ induced factor restricting *T. gondii* infection, the limited number of ISGs capable of restricting *T. gondii* infection when individually expressed suggests that IFNγ-mediated immunity to *T. gondii* infection is a complex, multifactorial process.

*For correspondence:
sibley@wustl.edu

## Editor's evaluation

*Toxoplasma gondii* is a widespread parasite of warm-blooded animals, with estimates suggesting 2 billion people are currently and chronically infected with this pathogen. Many questions remain as to how humans control and eliminate *T. gondii* following infection. In this manuscript, Rinkenberger et al. reveal a previously unidentified and understudied host factor, RARRES3 that promotes cell autonomous control of *T. gondii* in human cells by mediating pathogen expulsion from infected cells.

## Introduction

*Toxoplasma gondii* infection is common in humans, and while typically self-limiting in immunocompetent individuals, it can be severe in congenital toxoplasmosis and in immunodeficient individuals (*Furtado et al., 2011*). Additionally, *T. gondii* is a leading cause of infectious retinochoroiditis (*Weiss and Dubey, 2009*). Eye disease is most prominent in Latin America and Africa with approximately one-third of uveitis cases being attributed to ocular toxoplasmosis (*Furtado et al., 2013*). Although mechanisms of immune control are well studied in the mouse, they are less well understood in humans with currently known mechanisms of restriction observed in a cell type-dependent manner (*Fisch et al., 2019b*).

In response to pathogen infection, host cells express and secrete interferons (IFNs), which signal in an autocrine or paracrine manner through IFN receptors to induce factors designed to block infection.

Interferons fall into three categories: type I (including IFNα and IFNβ), type II (IFNγ), and type III (IFN $\lambda$ 1–4). In general, receptors for type I and type II IFN are ubiquitously expressed whereas type III IFN sensitivity is restricted to epithelial barriers. Conventionally, IFN signaling involves the induction of interferon-stimulated genes (ISGs) involved in host defense via JAK-STAT-mediated signaling (*Alspach et al., 2019*; *Lazear et al., 2019*; *Schoggins, 2019*). Although IFN upregulates the expression of many genes, induction of ISGs is only semi-conserved between cell types, with many ISGs being cell type dependent (*Schoggins, 2019*). Variability in ISG expression is perhaps due to the induction of noncanonical IFN signaling pathways that have been observed in a cell type-dependent manner (*Van Boxel-Dezaire and Stark, 2007*; *van Boxel-Dezaire et al., 2006*). To identify and study the functions of this diverse set of genes, a wide variety of approaches have been utilized including ectopic overexpression, siRNA-mediated knockdown, and more recently CRISPR-Cas9 screening approaches (*Schoggins, 2019*). For the purpose of our study, several previous overexpression screens have been informative: an ectopic expression-based screen of type II IFN induced genes developed by *Abrams et al., 2020*, and the prior screen developed by *Schoggins et al., 2011*, which focused on type I IFN induced genes. These screens utilized a lentiviral-based expression cassette to express a curated library of commonly expressed ISGs in a one gene per well format. Abrams et al., successfully used this approach to identify novel type II IFN induced genes that impact *Listeria monocytogenes* infection while the screen developed by Schoggins et al., has been used to identify many ISGs impacting a broad range of pathogens, including both bacteria and viruses (*Abrams et al., 2020*; *Schoggins et al., 2011*; *Schoggins et al., 2014*; *Perelman et al., 2016*). A similar ectopic expression-based screen utilized a large cDNA library to search for enhancers of STAT1-mediated transcription in *T. gondii* infected cells and successfully identified the orphan nuclear receptor TLX as an enhancer of STAT1-mediated transcription (*Beiting et al., 2015*).

Interferon gamma (IFNγ) has been known since the 1980s to be expressed during *T. gondii* infection and to be critical for restricting infection in mice (*Suzuki et al., 1988*; *McCabe et al., 1984*; *Shirahata and Shimizu, 1980*). IFNγ-mediated restriction has been attributed to the expression of a group of ISGs, including immunity-related GTPases (IRGs) and guanylate binding proteins (GBPs) (*Gazzinelli et al., 2014*; *MacMicking, 2012*). IRGs and GBPs are recruited to the parasitophorous vacuole membrane (PVM) resulting in a loss of membrane integrity and parasite death (*Ling et al., 2006*; *Martens et al., 2005*; *Yamamoto et al., 2012*). As a defense, type I and type II strains of *T. gondii* express the Ser/Thr kinase ROP18, which phosphorylates and inactivates IRG proteins to prevent PVM damage (*Hunter and Sibley, 2012*; *Mukhopadhyay et al., 2020*). Additionally, increased nitric oxide production due to induction of inducible nitric oxide synthase (iNOS) has also been shown to play a role in restricting *T. gondii* infection in mice in vivo and in vitro (*Khan et al., 1997*; *Adams et al., 1990*; *Meisel et al., 2011*).

IFNγ is similarly important for *T. gondii* restriction in human cells in vitro and IFNγ expression has been shown to correlate with disease severity in vivo (*Pfefferkorn et al., 1986*; *Meira et al., 2014*). In contrast to the situation in mouse, the reported mechanisms underlying IFNγ-mediated *T. gondii* restriction in humans tend to be cell-type specific. However, it is important to note here that many of the studies in mice have been conducted in primary cells whereas human studies have mostly relied on immortalized cell lines. This could also be an important factor in the differences between restriction mechanisms observed in mice and humans. Humans possess one truncated IRG that likely lacks GTPase activity and only one nontruncated IRG that is not IFNγ or infection inducible (*Bekpen et al., 2005*). Hence, it is unlikely IRGs play a role in human resistance to *T. gondii* infection. Although human GBP1 has been shown to restrict *T. gondii* infection, it does so in a cell type-dependent manner. GBP1 restricts infection in IFNγ-treated human mesenchymal stromal cells (MSCs) and lung epithelial cells (i.e., A549 cells) but not myeloid-derived cells (i.e., HAP1 cells) (*Qin et al., 2017*; *Ohshima et al., 2014*; *Johnston et al., 2016*). GBP1 is also implicated in an inflammasome pathway that results in host cell death in human macrophages following *T. gondii* infection (*Fisch et al., 2019a*). Additionally, GBP5 has been shown to play a role in the clearance of *T. gondii* from human macrophages in vitro, albeit in an IFNγ independent manner (*Matta et al., 2018*). Humans also possess an ISG15-dependent, IFNγ inducible, noncanonical autophagy (ATG) pathway that restricts *T. gondii* growth in human cervical adenocarcinoma cells (HeLa cells) and A549 cells (*Bhushan et al., 2020*; *Selleck et al., 2015*). A similar noncanonical ATG dependent pathway has been reported in human umbilical vein epithelial cells (HUVECs), although it differs slightly in culminating in lysosome fusion (*Clough*

*et al., 2016*). Finally, indoleamine 2,3-dioxygenase (IDO1) has been shown in vitro to restrict *T. gondii* growth by limiting L-tryptophan availability in human fibroblasts and monocyte-derived macrophages as well as in human-derived cell lines of myeloid, foreskin, liver, or cervix origin (e.g., HAP1s, HFFs, Huh7s, and HeLas) but not in cells lines originating from mesenchymal stem cells, the large intestine, or the umbilical endothelium (e.g., MSCs, CaCO$_2$s, or HUVECs) (*Qin et al., 2017*; *Bando et al., 2018*; *Pfefferkorn, 1984*; *Woodman et al., 1991*; *Dimier and Bout, 1997*; *Schmitz et al., 1989*).

Collectively, the previous studies examining IFNγ-mediated growth restriction in human cells support a model where different mechanisms make variable contributions in distinct lineages. However, the known pathways for restriction only cover a small fraction of the genes that are normally upregulated in different human cell types following treatment with IFNγ (*Schoggins, 2019*; *Rusinova et al., 2013*). Hence, there may be additional control mechanisms not yet defined, including either those that are lineage-specific or that operate globally in all cell types. To explore this hypothesis, we screened a library of 414 IFNγ induced ISGs in a one gene per well format to attempt to identify novel human factors with the ability to restrict *T. gondii* infection.

## Results

To identify novel ISGs that impact *T. gondii* infection, we employed a library of 414 IFNγ induced ISGs cloned into a lentiviral expression cassette co-expressing tagRFP, as previously described by *Abrams et al., 2020*. To screen for ISGs that restrict *T. gondii* infection, we developed a high-throughput method to quantitatively measure infection of GFP-expressing type III strain CTG parasites using auto-mated microscopy. CTG is a type III strain of *T. gondii* that is avirulent in mice and more susceptible to IFNγ-mediated restriction than other strains thus making it ideal for identifying individual ISGs which can restrict infection (*Khaminets et al., 2010*; *Boothroyd and Grigg, 2002*; *Saeij et al., 2006*). A549 lung epithelial cells were infected with CTG-GFP and the size of individual parasitophorous vacuoles (PVs), the number of vacuoles per field, and the percentage of vacuoles with a size consistent with containing ≥8 parasites were determined after 36 hr of culture (*Figure 1A–C*). A549 cells were used as they are highly permissive to transduction with our VSV-G pseudotyped lentivirus and readily infected by *T. gondii*. Cells were transduced with lentivirus in a one gene per well format and challenged with CTG-GFP (*Figure 1D*). Transduction efficiency was high for most ISGs with 86% (357/414) of ISGs expressed in at least 50% of the cell population and 50% of ISGs (205/414) expressed in 90% of the cell population (*Supplementary file 1*). Using this approach, we found three ISGs that restricted infection: IRF1, TRIM31, and RARRES3 (*Figure 1E*, *Supplementary file 1*). All hits were identified as significant by a two-way ANOVA (p<0.0001). Interestingly, six ISGs significantly promoted parasite growth, including IL7R, IFITM1, MX1, DHX58, RNF19B, and CNDP2. For the purposes of this study, we were interested in investigating ISGs that restricted infection and did not further validate or study any ISGs that promoted parasite growth. We found it curious that IDO1 was not identified by this screen considering its significant impact on infection observed in some but not all cell lines (*Qin et al., 2017*; *Bando et al., 2018*; *Pfefferkorn, 1984*; *Woodman et al., 1991*; *Dimier and Bout, 1997*; *Schmitz et al., 1989*). There have been no previous reports as to the role of IDO1 during infection in A549s and as such we generated an IDO1 deficient A549 cell line and challenged with CTG-GFP in the pres-ence of IFNγ. Consistent with the results of our screen, IDO1 deficiency did not impair IFNγ-mediated restriction of CTG-GFP (*Figure 1F*). It is worth noting here that the concentration of tryptophan used in the media for these experiments (16 μg/ml) was higher than what has been used previously to observe IDO1-mediated infection restriction, which may prevent IDO1-mediated depletion of trypto-phan (*Pfefferkorn, 1984*). However, in the present study, a much higher dose of IFNγ was also used to induce strong ISG expression and this might be expected to overcome increased tryptophan levels. Collectively, our results suggest that IDO1 does not play a major role in the restriction of *T. gondii* in A549 cells under the experimental conditions used in this study.

### Validation of screen hits

Cells ectopically expressing IRF1, TRIM31, and RARRES3 expanded normally and showed similar viability compared to a luciferase control when stained with the live-dead stain SYTOX green, suggesting that the overexpression of these genes is not cytotoxic (*Figure 2A–B*). To confirm that these genes restrict *T. gondii* infection, we ectopically expressed these genes in A549 cells and

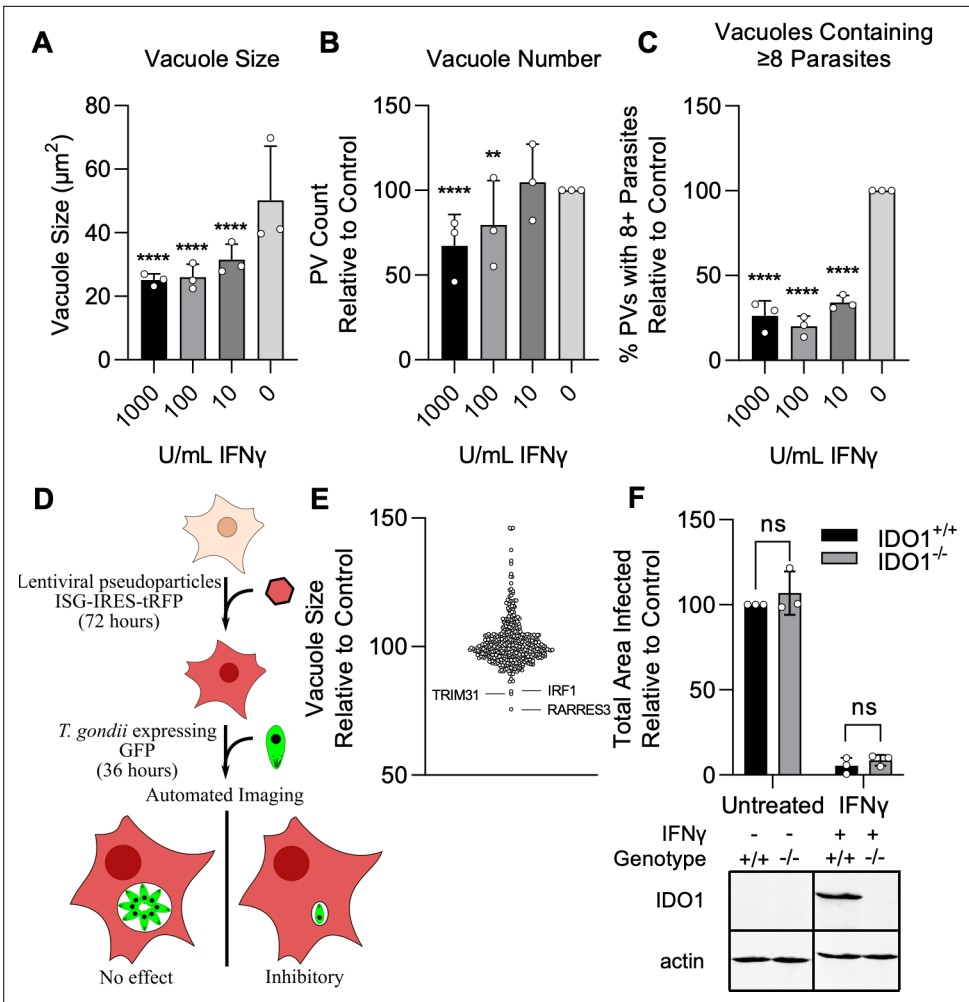

**Figure 1.** Screen for interferon-stimulated genes (ISGs) impacting *Toxoplasma gondii* infection. A549 cells were treated with indicated concentrations of IFNγ for 24 hr and subsequently infected with the type III strain CTG expressing GFP (CTG-GFP) for 36 hr. (**A–C**) Cells were fixed, stained with anti-GFP and anti-RFP antibodies, and imaged using a Cytation3 Imager. (**A–C**) Average parasitophorous vacuole (PV) size (**A**), PVs per field (**B**), and the percentage of vacuoles containing ≥8 parasites (**C**) was quantitated for data from 36 hr image sets. (**D**) Illustration of the method used to conduct the screen presented in (**E**). (**E**) A549 cells were transduced with a lentiviral expression cassette co-transcriptionally expressing tagRFP and an ISG of interest in a one gene per well format. After 72 hr, cells were infected with CTG-GFP for 36 hr, fixed, stained with anti-GFP and anti-RFP antibodies, and imaged with a Cytation3 Imager. Statistical significance was determined using a two-way ANOVA with Tukey's test for post hoc analysis. ISGs enhancing or restricting infection >20% relative to control with p<0.0001 were classified as hits. Hits are shown in red and labeled. (**F**) WT and IDO1$^{-/-}$ A549 cells were infected with CTG-GFP for 96 hr. Cells were fixed, stained with anti-SAG1 antibody, and imaged with a Cytation3 Imager. Average total infected area per well is shown. Loss of IDO1 in IDO1$^{-/-}$ A549 cells was confirmed via western blot. Briefly, cells were treated with or without 1000 U/ml IFNγ for 24 hr before samples were harvested and IDO1 expression was determined. (**A–C, F**) Data represent the deviation of three biological replicates conducted in technical triplicate. (**E**) Data represent deviation of two biological replicates conducted in technical duplicate. Statistical significance was determined using two-way ANOVA with Tukey's test for post hoc analysis. ns, not significant; p>0.05, **p<0.01, ****p<0.0001.

The online version of this article includes the following source data for figure 1:

**Source data 1.** Vacuole size quantitation following CTG-GFP infection of IFN pretreated A549 cells.

**Source data 2.** Vacuole number quantitation following CTG-GFP infection of IFN pretreated A549 cells.

**Source data 3.** Quantitation of the percentage of vacuoles containing eight or more parasites following CTG-GFP infection of IFN pretreated A549 cells.

**Source data 4.** Numerical value summary for the results of the ISG screen.

*Figure 1 continued on next page*

*Figure 1 continued*

**Source data 5.** Quantitation of the total image area infected after CTG-GFP infection of WT or IDO1$^{-/-}$ A549 cells.

**Source data 6.** Contrast enhanced and labeled actin western blot.

**Source data 7.** Unmodified actin western blot.

**Source data 8.** Contrast enhanced and labeled IDO1 western blot.

**Source data 9.** Unmodified IDO1 western blot.

challenged with CTG-GFP for 36 or 96 hr. RARRES3 and IRF1 ectopic expression resulted in a reduction in average vacuole size at 36 hr (*Figure 2C–D*). At 96 hr, the total area infected per well was significantly lower for RARRES3 and IRF1 expressing cells compared to control as was the average size of infection foci (*Figure 2E–F*). TRIM31 had no significant effect on infection in validation experiments and it was not studied further. IRF1 has a known role in amplifying IFNγ-mediated transcription, and its role relative to IFNγ is further explored below. RARRES3 is a small, 18.2 kDa single domain protein in the HRASLS subfamily (*Mardian et al., 2015*). RARRES33 has been shown to inhibit immunoproteasome expression in cancer cells (*Anderson et al., 2017*). It displays phospholipase A1/2 activity in vitro and has been shown to suppress Ras signaling and promote apoptosis (*Uyama et al., 2009*; *Tsai et al., 2007*; *Han et al., 2010*). Although RARRES3 was described by a previous screen to be antiviral (*Schoggins et al., 2014*), it was not studied further and little is known about its involvement in immunity to other pathogens. In studies described below, we explore its role in restricting the growth of intracellular *T.gondii*.

## IRF1 versus IFNγ induced genes

In mice, Irf1 deficiency has been shown to result in increased susceptibility to *T. gondii* infection (*Khan et al., 1996*), consistent with the secondary induction of a broad set of ISGs downstream of IFN dependent STAT-mediated gene expression (*Feng et al., 2021*). However, it is unknown what subset of IFNγ-induced genes are regulated by IRF1 in A549 cells. To define these two gene sets, we ectopically expressed IRF1 or luciferase control in A549 cells, treated a subset of control cells with IFNγ, and analyzed transcriptional changes by RNA-seq. For both IFNγ and IRF1, the majority of changes were due to upregulation, and we focused our analysis on these genes (*Figure 3A and B*). We identified 160 genes upregulated by IRF1 and 380 genes upregulated by IFNγ in A549 cells (FDR≤0.05, twofold) (*Supplementary file 2*). When we compared these gene lists with the ISG library with which we challenged *T. gondii* infection, we found that 41.1% (86/160) of IFNγ induced genes and 53.8% (156/380) of IRF1 induced genes were represented by the library (*Supplementary file 3*). Notably, strongly induced ISGs were more commonly represented in the ISG library with 69% of the top 100 strongest induced genes by IFNγ and 67% of those induced by IRF1 being represented (*Figure 3C*, *Supplementary file 3*). Gene ontology (GO) analysis for the lists of IRF1 and IFNγ induced genes revealed induction of very similar processes that were grouped into IFN signaling, immune response regulation, host defense, and antigen presentation (*Figure 3D–E*). Interestingly, RARRES3 was strongly induced by both IRF1 and IFNγ treatment. This is consistent with the previously reported finding that IRF1 expression strongly induces RARRES3 expression in human hepatoma and skin fibroblast derived cell lines (*Schoggins et al., 2011*).

## RARRES3 does not affect immune signaling

To determine if the observed reduction in infection with RARRES3 overexpression might also be due to induction of interferon expression and downstream ISG induction, we used CRISPR/Cas9-mediated gene editing to generate a STAT1$^{-/-}$ A549 cell line (*Figure 4A–B*). We ectopically expressed RARRES3 in these cells and subsequently infected with CTG-GFP parasites. The same phenotypes were observed with RARRES3 ectopic expression irrespective of the presence of STAT1, suggesting that decreased infection on overexpression of RARRES3 is not due to induction of IFNγ signaling (*Figure 4C–F*). Interestingly, we noticed that RARRES3 ectopic expression resulted in a modest increase in the number of PVs observed at 36 hr in WT cells and was slightly more pronounced in STAT1$^{-/-}$ cells (*Figures 2C and 4C*). We further tested if ectopic expression of RARRES3 impacted NF-κB or interferon signaling by using κB-, ISRE-, and GAS-luciferase reporter cell lines. RARRES3 did not significantly impact luciferase

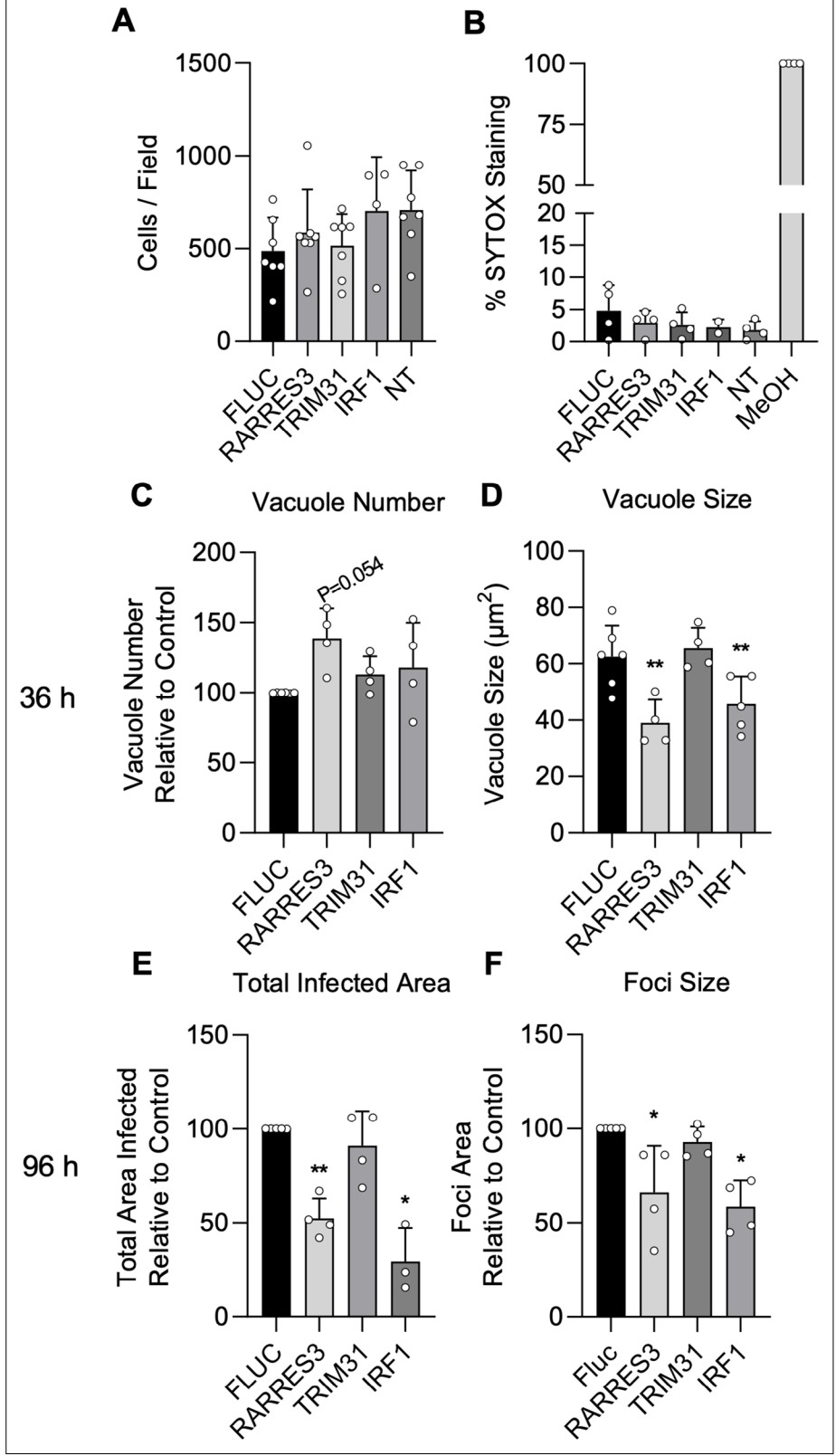

**Figure 2.** IRF1 and RARRES3 restrict *Toxoplasma* infection. (**A, B**) Wild-type (WT) A549 cells were either not transduced (NT) or transduced with TRIP.RARRES3 or TRIP.FLUC control and split 48 hr later. After 60 hr, cells were stained with Hoechst 33342, SYTOX green, and imaged with a Cytation3 Imager. As a positive control, cells were permeabilized by treatment with methanol (MeOH) for 5 min prior to staining. Average cell number (**A**) and the

*Figure 2 continued on next page*

*Figure 2 continued*

percentage of SYTOX staining cells (**B**) were determined. (**C–F**) WT A549 cells were transduced with TRIP.RARRES3 or TRIP.FLUC control and infected 72 hr later with CTG-GFP for 36 (**C, D**) or 96 (**E, F**) hr. Cells were fixed, stained with anti-GFP and anti-RFP antibodies, and imaged using a Cytation3 Imager. Average PV number per field (**C**) and PV size (**D**) were quantitated for 36 hr infections while total area infected per sample (**E**) and average foci size (**F**) were quantitated for 96 hr infections. Data in (**A**) represent four to seven biological replicates conducted in technical triplicate. Data in (**B**) represent two to four biological replicates conducted in technical triplicate. Data in (**C–F**) represent three to four biological replicates conducted in technical triplicate. Statistical significance was determined using a Brown-Forsythe and Welch ANOVA (**A, B**) or a two-way ANOVA with Tukey's test for post hoc analysis (**C–F**). *$p \leq 0.05$, **$p < 0.01$.

The online version of this article includes the following source data for figure 2:

**Source data 1.** Cell counts per field for cells ectopically expressing ISG screen hits.

**Source data 2.** Percentage of A549s ectopically expressing ISG screen hits that stained positive for SYTOX Green.

**Source data 3.** Number of vacuoles counted after infection of A549s ectopically expressing ISG screen hits with CTG-GFP for 36 hr.

**Source data 4.** Quantitation of vacuole size after infection of A549s ectopically expressing ISG screen hits with CTG-GFP for 36 hr.

**Source data 5.** Quantitation of total area infected per sample after infection of A549s ectopically expressing ISG screen hits with CTG-GFP for 96 hr.

**Source data 6.** Quantitation of average infection foci size formed after infection of A549s ectopically expressing ISG screen hits with CTG-GFP for 96 hr.

---

expression for any of the reporters tested (*Figure 4G–J*). These findings suggest that RARRES3 does not modulate immune signaling pathways and instead plays a direct role in restricting infection.

## Catalytic activity is required for infection restriction and endogenous RARRES3 can restrict infection

As previously stated, RARRES3 displays phospholipase A1/2 activity in vitro and belongs to the H-RAS suppressor-like (HRASLA) subfamily (*Uyama et al., 2009*). Phospholipase and acyltransferase activities of HRASLS subfamily enzymes involve the temporary acylation of an active site cysteine residue (*Mardian et al., 2015*). To determine if the catalytic activity of RARRES3 was required for restriction of infection, we generated two active site point mutants (C113A, C113S) of RARRES3. Cells ectopically expressing mutant or WT RARRES3 were subsequently challenged with CTG-GFP. Although WT RARRES3 restricted infection, neither mutant was able to do so (*Figure 5A*). Notably, RARRES3 C113A ectopic expression slightly promoted infection. Analysis of protein expression by western blot indicated that both mutants were strongly expressed compared to WT (*Figure 5B*). These data suggest that the enzymatic activity of RARRES3 is necessary for restriction of infection. To determine if endogenously expressed RARRES3 impacts infection, we next used CRISPR/Cas9-mediated gene editing to generate a RARRES3$^{-/-}$ A549 cell line. Alternatively, cells were transduced with an expression cassette containing Cas9 and a previously used nontargeting sgRNA to serve as a negative control (*Doench et al., 2016*). RARRES3 deficiency did not alter the susceptibility of quiescent cells to infection (*Figure 5C–D*). However, RARRES3 deficiency partially alleviated IFNγ-mediated restriction of CTG-GFP infection and this deficiency was complemented with RARRES3 ectopic expression (*Figure 5C–D*). As previously mentioned, RNA-seq analysis showed that RARRES3 expression was strongly induced by both IRF1 and IFNγ. Since RARRES3 was the only ISG identified by our screen to restrict *T. gondii* infection, we wanted to determine if the impact of IRF1 on infection was solely due to upregulation of RARRES3 expression. To test this, we ectopically expressed IRF1 or luciferase control in WT and RARRES3$^{-/-}$ A549 cells and challenged them with CTG-GFP. However, loss of RARRES3 did not impact IRF1-mediated restriction of infection (*Figure 5E*). This suggests that either IRF1-mediated infection restriction is RARRES3 independent or involves multiple factors with RARRES3 playing a redundant role in the process of restricting growth. As previously mentioned, CTG is a type III strain of *T. gondii* that is less virulent in mice and more susceptible to IFNγ-mediated restriction than other strains (*Khaminets et al., 2010*; *Boothroyd and Grigg, 2002*; *Saeij et al., 2006*). We wanted to determine if RARRES3 also impacts other strains of *T. gondii*. To test this, we infected A549s ectopically

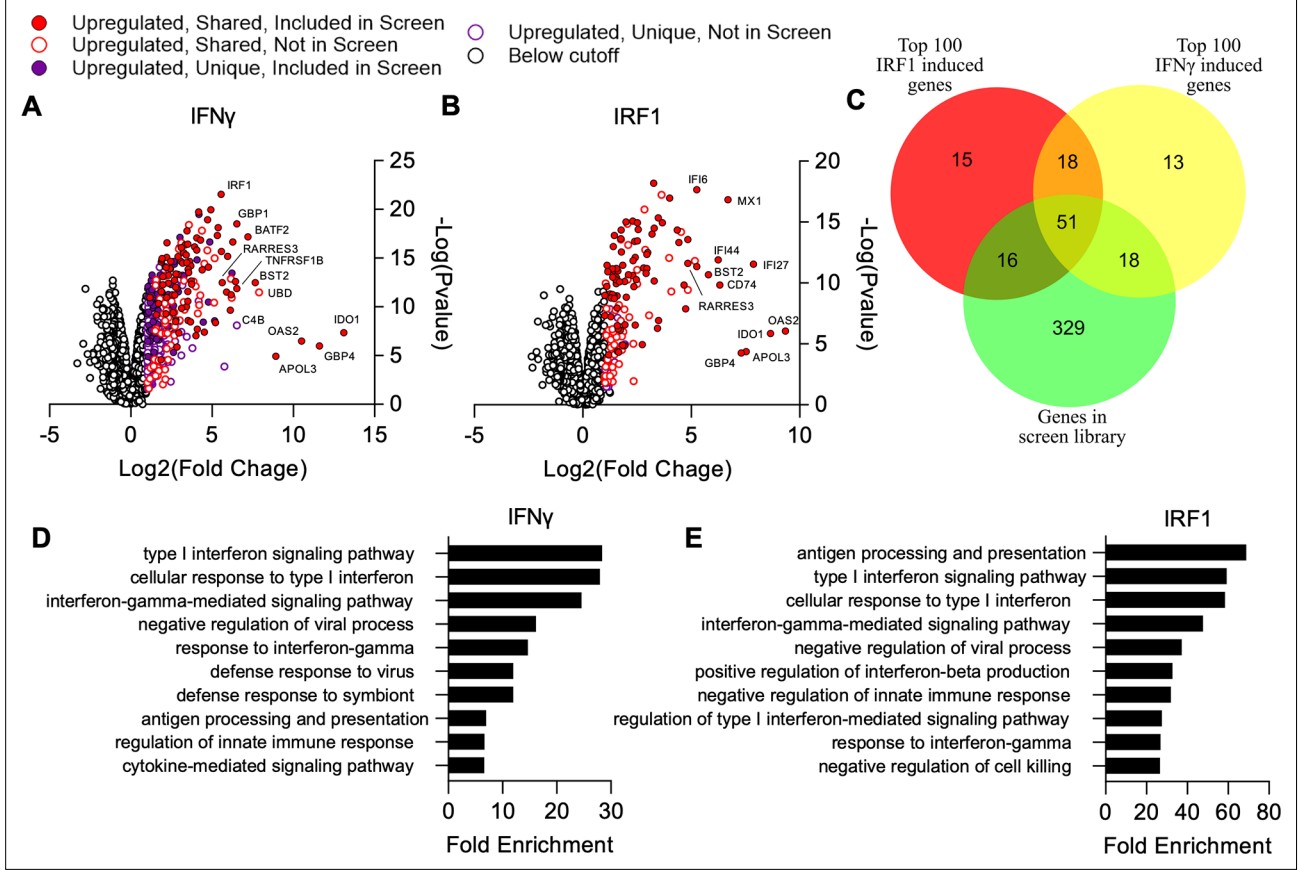

**Figure 3.** Comparison of genes induced by IRF1 and IFNγ in A549 cells. Cells were transduced with TRIP.IRF1 or TRIP.FLUC control lentivirus. Cells transduced with FLUC were further treated 72 hr later with or without 1000 U/ml IFNγ for 24 hr. All cell populations were subsequently harvested and analyzed by RNA-Seq. (**A, B**) Changes in gene expression relative to FLUC control expressing cells for cells treated with IFNγ (**A**) or ectopically expressing IRF1 (**B**). Genes upregulated in both IRF1 expressing and IFNγ treated cells are defined as 'Shared' while genes only upregulated in one of these two cell populations are defined as 'Unique.' (**C**) Comparison of genes induced ≥2-fold with a false discovery rate cutoff of 0.05 by each condition and their overlap with the ISG library used in the screen described in *Figure 1*. (**D–E**) Lists of induced genes were analyzed with PANTHER gene ontology analysis. The top 10 most enriched processes amongst genes induced by IFNγ (**D**) and IRF1 (**E**) are shown. Redundant terms were excluded from these lists with only the most enriched version of each term remaining.

The online version of this article includes the following source data for figure 3:

**Source data 1.** Differential expression of genes after IFNγ treatment relative to mock control.

**Source data 2.** Differential expression of genes in IRF1 overexpressing cells relative to FLUC expressing control cells.

**Source data 3.** Gene ontology term enrichment for genes significantly induced by IFNγ treatment.

**Source data 4.** Gene ontology term enrichment for genes significantly induced by IRF1 overexpression.

expressing RARRES3 with GFP expressing RH88 (type I) and Me49 (type II). However, RARRES3 had no impact on the infection of either of these two strains (*Figure 5—figure supplement 1*). Differential expression or polymorphic virulence factors may explain the disparity in susceptibility to RARRES3-mediated restriction between strains (*Saeij et al., 2006*; *Rosowski et al., 2011*).

## RARRES3 promotes premature egress

We subsequently revisited our previous finding that infection in RARRES3 ectopically expressing cells results in more, smaller PVs at 36 hr than control. This result suggested to us that RARRES3 might be promoting premature egress of the parasite. To test this possibility, we infected A549 cells ectopically expressing RARRES3 with CTG-GFP parasites and measured lactate dehydrogenase (LDH) release (*Figure 6A*). RARRES3 ectopic expression resulted in increased LDH release in CTG-infected cells compared to control cells (*Figure 6A*). We next sought to differentiate if the observed LDH release was due to protein kinase G (PKG) dependent, active parasite egress, or a form of induced cell

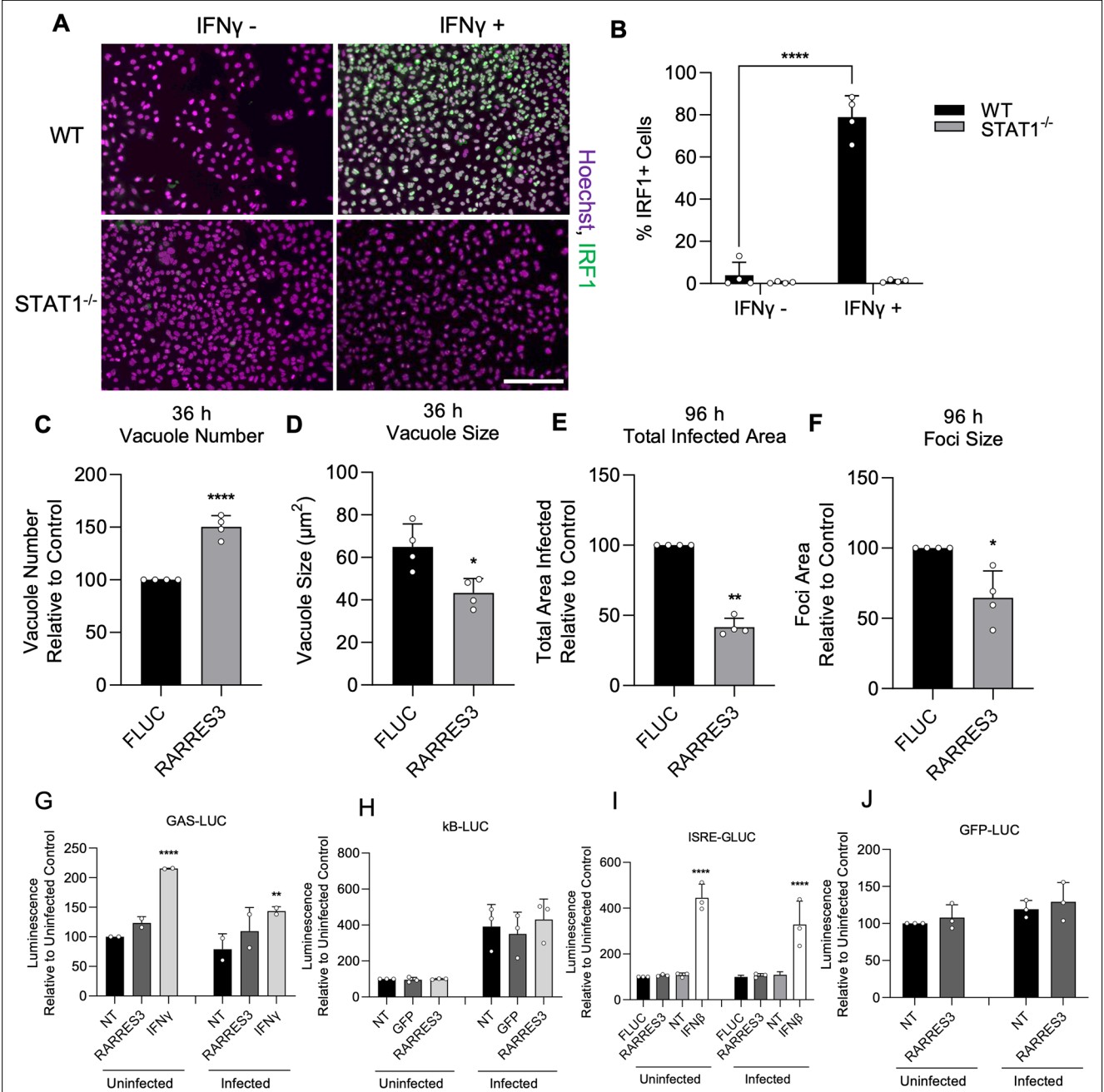

**Figure 4.** RARRES3 restricts *Toxoplasma* infection in a STAT1 independent manner. To determine if restriction of *T. gondii* growth was STAT1 dependent, STAT1⁻/⁻ A549 cells were generated. To confirm complete insensitivity to interferon treatment, WT or STAT1⁻/⁻ A549 cells were treated with or without 4000 U/ml IFNγ for 6 hr, fixed, stained with anti-IRF1 antibodies, and imaged with a Cytation3 Imager. (**A**) Representative images and (**B**) quantitation are shown. Scale bar=50 µm. (**C–F**) STAT1⁻/⁻ A549 cells were transduced with TRIP.RARRES3 or TRIP.FLUC control and infected 72 hr later with CTG-GFP for 36 (**C, D**) or 96 (**E, F**) hr. Cells were fixed, stained with anti-GFP and anti-RFP antibodies, and imaged using a Cytation3 Imager. Average PV number per field (**C**) and PV size (**D**) were quantitated for 36 hr infections while total area infected per sample (**E**) and average foci size (**F**) were quantitated for 96 hr infections. HeLa reporter cell lines expressing GAS-LUC (**G**), kB-LUC (**H**), ISRE-GLUC (**I**), and GFP-LUC (**J**) were either not transduced (NT) or transduced with TRIP.RARRES3, TRIP.FLUC, or TRIP.GFP. After 72 hr, cells were mock treated or treated with 100 U/ml IFNβ or IFNγ as indicated and infected with CTG-GFP for 36 hr. Cells were harvested for luciferase assay. Data in (**B**) represent means ± SD of four biological replicates conducted in technical duplicate. Data in (**C–F**) represent means ± standard deviation of four biological replicates conducted in technical triplicate. Data represent means ± SD of two (**G**) or three (**H–I**) biological replicates conducted in technical duplicate. Statistical significance was determined using two-way ANOVA with Tukey's test for post hoc analysis except for (**D**) where Mann-Whitney's U-test was used. *p≤0.05, **p<0.01, ***p<0.001, ****p<0.0001.

The online version of this article includes the following source data for figure 4:

*Figure 4 continued on next page*

*Figure 4 continued*

**Source data 1.** Quantitation of the percentage of IRF1 staining cells after IFNγ treatment of WT or STAT1$^{-/-}$ A549 cells.

**Source data 2.** Quantitation of the number of vacuoles formed following CTG-GFP infection of RARRES3 or FLUC ectopically expressing STAT1$^{-/-}$ A549 cells.

**Source data 3.** Vacuole size quantitation following CTG-GFP infection of RARRES3 or FLUC ectopically expressing STAT1$^{-/-}$ A549 cells.

**Source data 4.** Quantitation of total infected area per sample after infection of STAT1$^{-/-}$ A549s ectopically expressing RARRES3 or FLUC with CTG-GFP for 96 hr.

**Source data 5.** Quantitation of average infection foci size formed after infection of STAT1$^{-/-}$ A549s ectopically expressing RARRES3 or FLUC with CTG-GFP for 96 hr.

**Source data 6.** Luminescence values from luciferase assays conducted on lysates from cells expressing a GAS-LUC reporter and RARRES3.

**Source data 7.** Luminescence values from luciferase assays conducted on lysates from cells expressing a $\kappa$ B-LUC reporter and RARRES3.

**Source data 8.** Luminescence values from luciferase assays conducted on cell supernatant from cells expressing an ISRE-GLUC reporter and RARRES3.

**Source data 9.** Luminescence values from luciferase assays conducted on lysates from cells expressing a GFP-LUC reporter control and RARRES3.

death. To differentiate between these two hypotheses, we treated cells during infection with a trisubstituted pyrrole *T. gondii* protein kinase G inhibitor known as Compound 1 (*Gurnett et al., 2002*; *Donald et al., 2006*). Compound 1 is a potent inhibitor of parasite egress and has also been shown to promote differentiation from tachyzoites to bradyzoites, significantly slowing parasite growth thus also delaying or preventing egress (*Nare et al., 2002*; *Lourido et al., 2012*; *Radke et al., 2006*). Treatment with Compound 1 blocked the LDH release observed during infection and compound 1 treated cells did not stain with propidium iodide, suggesting that host cell death as measured by LDH release was due to parasite egress (*Figure 6B*, *Figure 6—figure supplement 1A*). To further confirm that RARRES3 induces premature egress, we directly measured the kinetics of CTG-GFP egress from RARRES3 or FLUC control expressing cells via time-lapse imaging of live cells by video microscopy. Parasites egressed significantly sooner in RARRES3 expressing cells compared to control (two-way ANOVA, p<0.0001), further confirming that RARRES3 induces premature parasite egress (*Figure 6C*, *Videos 1–2*).

In addition to increased parasite egress, we observed a reduction in host cell number during infection in cells ectopically expressing RARRES3 compared to control at 36 hr postinfection (*Figure 6D*). Similar to above, this phenotype was blocked by Compound 1 addition. A common trigger of premature egress is the induction of cell death pathways, of which there is a diversity of types controlled by different mechanisms (*Yao et al., 2017*; *Persson et al., 2007*; *Niedelman et al., 2013*; *Rosenberg and Sibley, 2021*). If induction of host cell death is the trigger for premature egress, we would expect to observe host cell death even when parasite egress is blocked. Hence, we were curious if extending this timepoint further would reveal host cell death even with blockage of parasite egress by Compound 1. However, Compound 1 addition during infection resulted in no significant increase in LDH release or propidium iodide staining even 72 hr after infection (*Figure 6E*, *Figure 6—figure supplement 1B*). Meanwhile, in untreated, infected cultures, the cell monolayer was nearly completely lysed and LDH activity was elevated in the supernatant (*Figure 6E*, *Figure 6—figure supplement 1C*). Further, we did not observe signs of death such as cell rounding, blebbing, or loss of nuclear integrity prior to parasite egress during live imaging (*Videos 1–2*). Collectively, these findings suggest that induction of cell death by RARRES3 is not the trigger for promotion of parasite egress. To further test this idea, we treated cells with a panel of established inhibitors of known cell death pathways during infection either individually or in combination as indicated (*Figure 6F*). These inhibitors included the pan-caspase inhibitor Z-VAD-FMK, which blocks apoptosis; the necroptosis inhibitors GSK'963, GSK'872, and necrosulfonamide (NSA), which block RIP1, RIP3, and MLKL activity, respectively; and the pyroptosis inhibitor Z-YVAD-FMK, which blocks caspase 1. No single drug or combination thereof was capable of inhibiting LDH release during infection except for GSK'872, which partially prevented LDH release in both RARRES3 ectopically expressing and control cells. Overall, we conclude from these experiments that it is unlikely that overexpression of RARRES3 induces cell death and therefore it must trigger premature egress by some other means.

A similar premature egress phenotype to that observed here has previously been identified in HFF cells (*Niedelman et al., 2013*). We hypothesized that RARRES3 might play a role in this process. To test this possibility, we ectopically expressed RARRES3 in HFF cells and infected with CTG-GFP parasites.

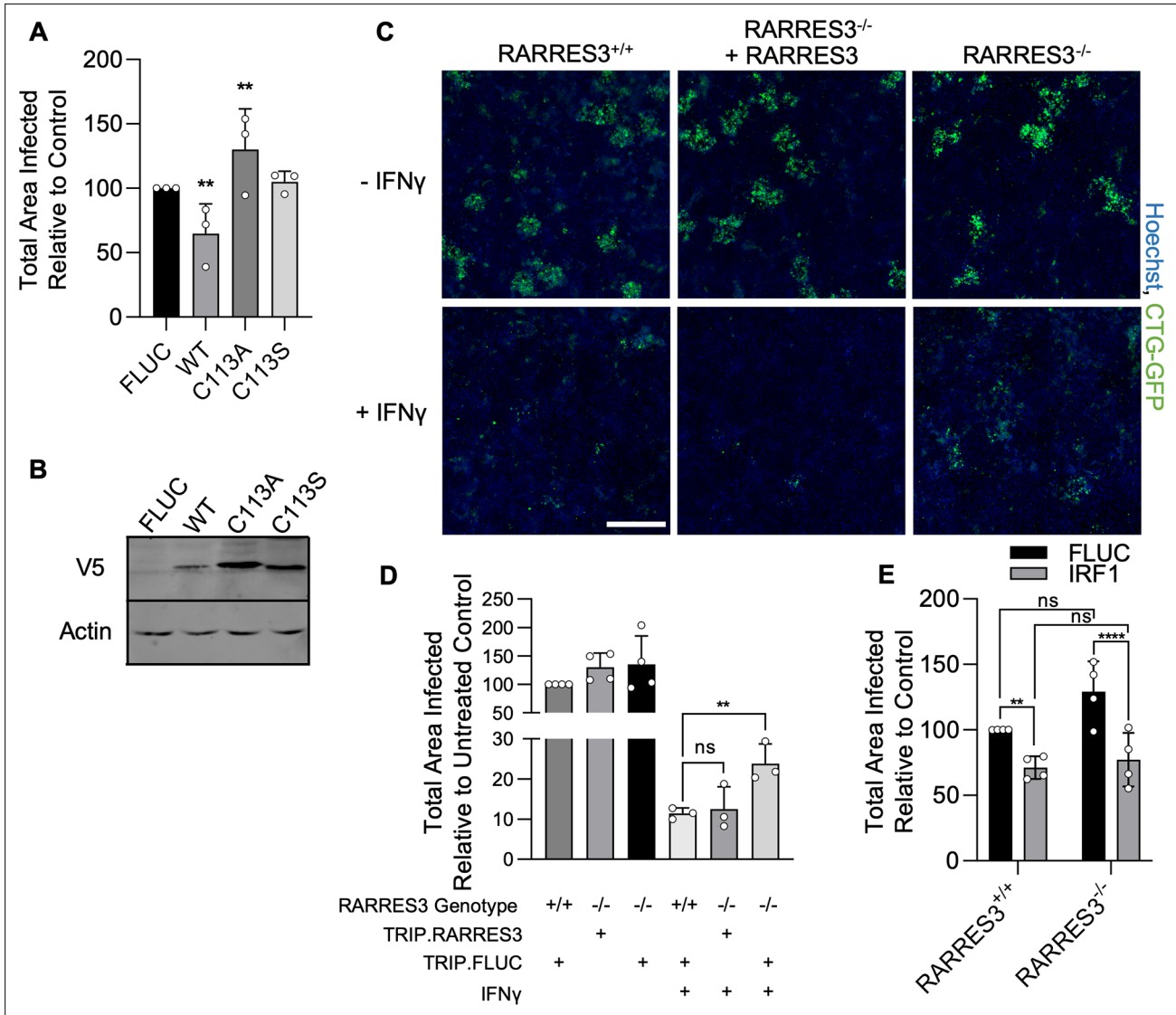

**Figure 5.** RARRES3 deficiency partially reverses IFNγ-mediated restriction of *Toxoplasma* infection. (**A, B**) A549s were transduced with V5 tagged WT RARRES3 or the catalytically inactive mutants C113A or C133S for 72 hr. (**A**) Cells were infected with CTG-GFP for 96 hr. Cells were harvested, stained with anti-GFP and anti-RFP antibodies, and imaged with a Cytation3 Imager. The total area infected per sample is shown. (**B**) Western blot from lysates prepared with above RARRES3 expressing cells. Membranes were probed with mouse anti-actin and mouse anti-V5 antibodies overnight followed by goat anti-mouse 680RD and imaged with a LI-COR Odyssey scanner. RARRES3⁻/⁻ A549s or wild-type cells transduced with a nontargeting CRISPR/Cas9 sgRNA were transduced with Cas9 resistant TRIP.RARRES3 or TRIP.FLUC as indicated. After 72 hr, cells were treated with or without 100 U/ml IFNγ for 24 hr as indicated and subsequently infected with CTG-GFP for 96 hr. Cells were harvested, stained with anti-GFP and anti-RFP antibodies, and imaged with a Cytation3 Imager. (**C**) Representative images and (**D**) quantitation are shown. Scale bar=500 µm. (**E**) RARRES3⁻/⁻ or RARRES3⁺/⁺ A549 cells transduced with a nontargeting CRISPR/Cas9 control sgRNA were transduced with TRIP.FLUC or TRIP.IRF1 derived lentivirus. After 72 hr, cells were infected with CTG-GFP for 96 hr, harvested, stained with anti-GFP and anti-RFP antibodies, and imaged with a Cytation3 Imager. Average total infected area per well is shown. Data represent means ± standard deviation of three (**A**) or four (**D, E**) biological replicates conducted in technical triplicate. Statistical significance was determined using two-way ANOVA with Tukey's test for post hoc analysis. ns, not significant; *p>0.05, **p<0.01, ****p<0.0001.

The online version of this article includes the following source data and figure supplement(s) for figure 5:

**Source data 1.** Quantitation of total infected area per sample after infection of A549s ectopically expressing WT RARRES3 and mutants with CTG-GFP for 96 hr.

**Source data 2.** Contrast enhanced and labeled actin western blot.

**Source data 3.** Unmodified actin western blot.

**Source data 4.** Contrast enhanced and labeled V5 tag western blot.

*Figure 5 continued on next page*

*Figure 5 continued*

**Source data 5.** Unmodified V5 tag western blot.

**Source data 6.** Quantitation of total infected area per sample after infection of IFNγ pretreated WT or RARRES3⁻/⁻ A549s with CTG-GFP for 96 hr.

**Source data 7.** Quantitation of total infected area per sample after infection of WT or RARRES3⁻/⁻ A549s ectopically expressing IRF1 with CTG-GFP for 96 hr.

**Figure supplement 1.** RARRES3 does not restrict infection of type I or II strains of *Toxoplasma gondii*.

**Figure supplement 1—source data 1.** Quantitation of total infected area per sample after infection of A549s ectopically expressing RARRES3 or FLUC with RH88-GFP.

**Figure supplement 1—source data 2.** Quantitation of average infection foci size formed after infection of A549s ectopically expressing RARRES3 or FLUC with RH88-GFP.

**Figure supplement 1—source data 3.** Quantitation of total infected area per sample after infection of A549s ectopically expressing RARRES3 or FLUC with Me49-GFP.

**Figure supplement 1—source data 4.** Quantitation of average infection foci size formed after infection of A549s ectopically expressing RARRES3 or FLUC with Me49-GFP.

RARRES3 ectopic expression was found to restrict CTG infection (*Figure 7A–B*). We next ablated RARRES3 expression in HFFs using CRISPR/Cas9-mediated gene editing. Loss of RARRES3 resulted in a partial reduction in IFNγ-dependent cell death in RARRES3⁻/⁻ HFF cells compared to control cells expressing Cas9 and a nontargeting sgRNA (*Figure 7C*). This finding indicates that RARRES3 plays a role in premature egress but is not the only factor involved in HFF cells. In line with this finding, RARRES3 deficiency in HFF cells partially abrogated IFNγ-mediated restriction of CTG-GFP infection (*Figure 7D*). Moreover, treatment with Compound 1 completely blocked cell death during infection, suggesting that cell death is caused by PKG-dependent parasite egress (*Figure 7E*, *Figure 6—figure supplement 1D-F*). Collectively, this data suggests that RARRES3 overexpression promotes premature parasite egress independently of host cell death pathways in multiple cell lines which presumably stunts parasite growth leading to reduced infection overall.

## Discussion

The major mechanisms of IFNγ-mediated immunity have been shown to be conserved in multiple cell types including embryonic fibroblasts, macrophages, and astrocytes suggesting high conservation of immune mechanisms against *T. gondii* in mice (*Ling et al., 2006*; *Martens et al., 2005*; *Yamamoto et al., 2012*; *Khan et al., 1997*; *Adams et al., 1990*; *Meisel et al., 2011*). However, the currently known restriction mechanisms in humans show dramatic differences between cell types with no common or widespread restriction mechanism being observed. On the other hand, of the hundreds of ISGs, the vast majority have never been studied in relation to *T. gondii*, leaving open the possibility of unidentified restriction mechanisms. Herein, we performed a screen of IFNγ induced ISGs to identify novel mechanisms of IFNγ-mediated immunity in humans. Our screen identified RARRES3 and IRF1 as ISGs restrictive to *T. gondii* infection in human cells. Further study revealed that RARRES3 induced premature egress of *T. gondii* from host cells in a cell death independent manner. Not unexpectedly, IRF1 induced genes were largely found to be a subset of those induced by IFNγ, including RARRES3. These findings suggest that IRF1 works by the collective action of multiple ISGs. The relative lack of individual ISGs that are active alone also suggests that control mechanisms effective against *T. gondii* rely on complexes of multiple ISGs or cellular proteins working in concert. Hence, the control of intracellular eukaryotic pathogens contrasts with that of viral pathogens, mirroring differences in their overall biological complexity.

We identified RARRES3 as an ISG promoting premature egress of type III strains of *T. gondii* in two different human cell lines. This phenotype occurs independently of triggering downstream immune signaling. A common cause of premature egress is the induction of host cell death pathways (*Yao et al., 2017*; *Persson et al., 2007*; *Niedelman et al., 2013*). And indeed, such a mechanism might be suggested due to the fact that RARRES3 ectopic expression has been previously shown to induce apoptosis (*Tsai et al., 2007*). However, in our system RARRES3 ectopic expression was not found to impact cell death. Additionally, basal levels of LDH release were not increased in cells ectopically expressing RARRES3. Finally, when we blocked cell death pathways individually or in tandem, we

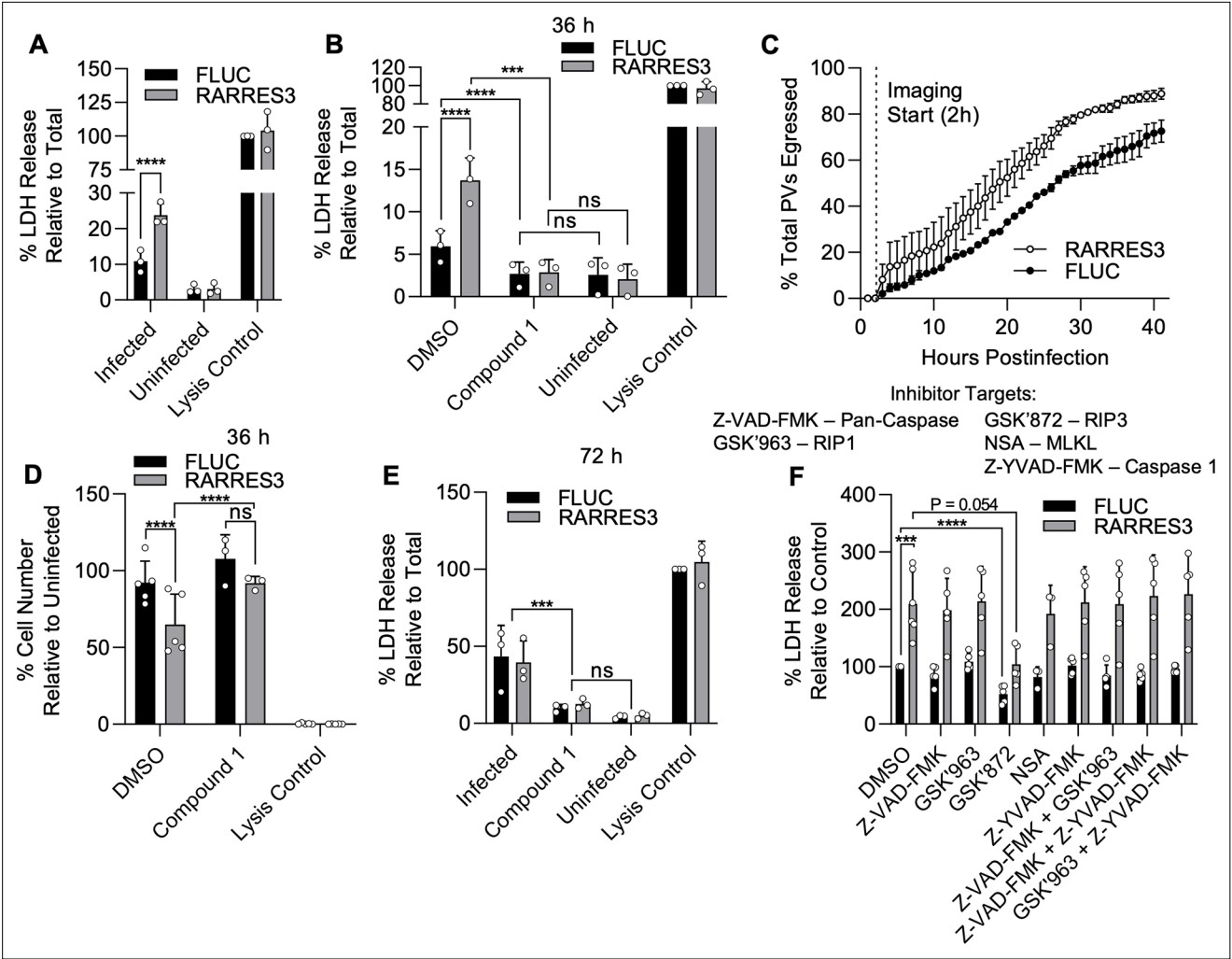

**Figure 6.** RARRES3 promotes premature egress of *Toxoplasma gondii*. A549 cells were transduced with TRIP.RARRES3 or TRIP.FLUC control and infected 72 hr later with CTG-GFP for 36 (**A, B, D, F**), 50 (**C**), or 72 (**E**) hr. Cells were treated with the cell death inhibitors Z-VAD-FMK (50 µM), GSK'963 (1 µM), GSK'872 (5 µM), NSA (10 µM), and Z-YVAD-FMK (10 µM) or the parasite egress inhibitor compound 1 (5 µM) as indicated during infection. (**A, B, E, F**) Cell supernatant was collected after infection and lactate dehydrogenase (LDH) activity was determined to measure cell lysis. As a control to measure maximal LDH release, cells were lysed before supernatant collection. (**C**) Live infection was imaged every 15 min starting 2 hr postinfection until 50 hr postinfection. Time of parasite egress was recorded for at least 100 PVs per condition per replicate. Percentage of total parasites egressed by the end of each hour is indicated. (**D**) Cells were fixed, stained with anti-RFP and anti-GFP antibodies, and imaged with a Cytation3 Imager. Average cells per field are shown. Data represent the means ± standard deviation (**A, B, D–F**) or standard error of the mean (**C**) of three to five biological replicates conducted in technical duplicate (**A, B, D–F**), or singlet (**C**). Statistical significance was determined using two-way ANOVA with Tukey's test for post hoc analysis. ns, not significant; P>0.05, *p≤0.05, ***p<0.001, ****p<0.0001.

The online version of this article includes the following source data and figure supplement(s) for figure 6:

**Source data 1.** Absorbance values for lactate dehydrogenase activity assays using supernatants from RARRES3 or FLUC ectopically expressing A549 cells infected with CTG-GFP for 36 hr.

**Source data 2.** Absorbance values for lactate dehydrogenase activity assays using supernatants from RARRES3 or FLUC ectopically expressing A549 cells infected with CTG-GFP for 36 hr in the presence of Compound 1.

**Source data 3.** Quantitation of the time of egress in hours for parasites egressing from RARRES3 or FLUC ectopically expressing A549s.

**Source data 4.** Cell count per field after infection of RARRES3 or FLUC ectopically expressing A549s with CTG-GFP in the presence of Compound 1.

**Source data 5.** Absorbance values for lactate dehydrogenase activity assays using supernatants from RARRES3 or FLUC ectopically expressing A549 cells infected with CTG-GFP for 72 hr in the presence of Compound 1.

**Source data 6.** Absorbance values for lactate dehydrogenase activity assays using supernatants from RARRES3 or FLUC ectopically expressing A549 cells infected with CTG-GFP in the presence of cell death inhibitors  .

**Figure supplement 1.** Compound 1 prevents host cell death during infection.

*Figure 6 continued on next page*

*Figure 6 continued*

**Figure supplement 1—source data 1.** Percentage of cells staining with propidium iodide after infection of RARRES3 or FLUC ectopically expressing A549s with CTG-GFP for 36 hr in the presence of Compound 1.

**Figure supplement 1—source data 2.** Percentage of cells staining with propidium iodide after infection of RARRES3 or FLUC ectopically expressing A549s with CTG-GFP for 72 hr in the presence of Compound 1.

**Figure supplement 1—source data 3.** Cell count per field after infection of RARRES3 or FLUC ectopically expressing A549s with CTG-GFP for 72 hr in the presence of Compound 1.

**Figure supplement 1—source data 4.** Cell count per field after infection of IFNγ pretreated HFFs with CTG-GFP for 36 hr in the presence of Compound 1.

**Figure supplement 1—source data 5.** Percentage of cells staining with propidium iodide after infection of IFNγ pretreated HFFs with CTG-GFP for 36 hr in the presence of Compound 1.

did not see a reduction in LDH. The one exception to this pattern was the reduced release of LDH following treatment with GSK'872. We suspect this result is due to RIP3 inhibition mediated induction of apoptosis as has been demonstrated previously with GSK'872 treatment (*Tenev et al., 2011*). Thus, following treatment with GSK'872 it is likely that infection is hindered by host cell apoptosis and LDH release is not observed since cell membranes remain largely intact. The early egress phenotype observed here was similar to an IFNγ-dependent premature parasite egress observed in HFF cells that was independent of known cell death pathways (*Niedelman et al., 2013*). We found that ablation of RARRES3 expression partially prevented the IFNγ and infection-dependent cell death phenotype in HFF cells. Furthermore, we observed that LDH release was blocked via Compound 1 mediated inhibition of PKG suggesting that RARRES3 overexpression leads to parasite egress resulting in host cell lysis. Although previous studies using CDPK3 inhibition concluded that blocking egress was not sufficient to prevent cell death, our findings differ from this conclusion, possibly due to the stronger role for PKG in controlling egress (*Lourido et al., 2012*). It is presently unclear how RARRES3 leads to premature egress, although it is possible that its phospholipase A1/A2 activity may alter cellular lipid pools that could induce premature egress consistent with the requirement for catalytic residues. As an example of the sensitivity of egress to membrane lipid constituents, increases in phosphatidic acid (*Bisio et al., 2019*), or the activity of lipolytic lecithin: cholesterol acyltransferase *Pszenny et al., 2016* have been shown to trigger egress of *T. gondii*.

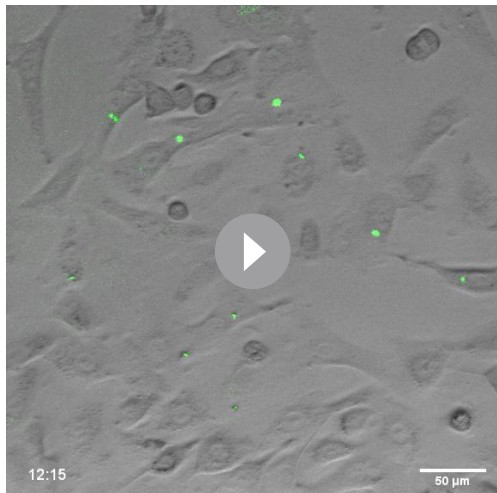

**Video 1.** Live imaging of FLUC control ectopically expressing A549s infected with CTG-GFP in 15 min intervals. A549 cells were transduced with TRIP.FLUC control and infected 72 hr later with CTG-GFP. Live infection was imaged every 15 min starting 12 hr postinfection until 50 hr postinfection.

https://elifesciences.org/articles/73137/figures#video1

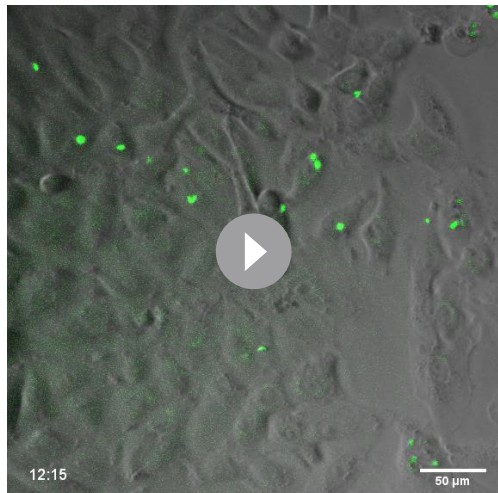

**Video 2.** Live imaging of RARRES3 ectopically expressing A549s infected with CTG-GFP in 15 min intervals. A549 cells were transduced with TRIP.RARRES3 and infected 72 hr later with CTG-GFP. Live infection was imaged every 15 min starting 12 hr postinfection until 50 hr postinfection.

https://elifesciences.org/articles/73137/figures#video2

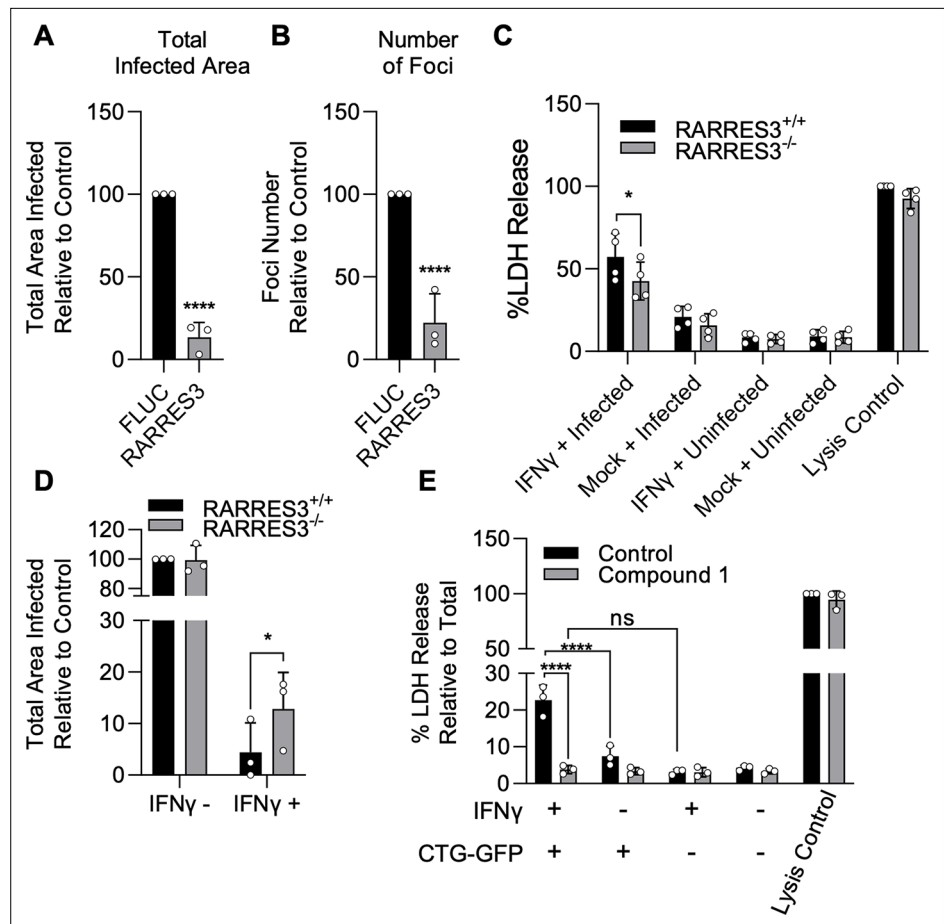

**Figure 7.** IFNγ-dependent host cell death during infection in HFFs is partially RARRES3 dependent. (**A, B**) HFF cells were transduced with TRIP.RARRES3 or TRIP.FLUC control and infected 72 hr later with CTG-GFP for 96 hr. Cells were fixed, stained with anti-GFP and anti-RFP antibodies, and imaged using a Cytation3 Imager. Total infected area per well (**A**) and average number of infection foci (**B**) are shown. (**C–D**) WT HFFs expressing a nontargeting sgRNA control or RARRES3 deficient HFFs were pretreated with or without 1000 U/ml IFNγ for 24 hr. (**C**) Cells were infected with CTG-GFP for 36 hr. Supernatant was collected and lactate dehydrogenase (LDH) activity was determined. As a control to measure maximal LDH release, cells were lysed before supernatant collection. (**D**) Cells were infected with CTG-GFP for 96 hr. Samples were treated as in (**A**). Total infected area per sample is shown. (**E**) HFFs were infected with CTG-GFP for 36 hr in the presence or absence of 5 µM Compound 1. Supernatant was collected and LDH activity was determined. Data represent means ± standard deviation of three (**A, B, D, E**) or four (**C**) biological replicates conducted in technical duplicate (**A, B, D, E**) or singlet (**C**). Statistical significance was determined using two-way ANOVA with Tukey's test for post hoc analysis. *$p \leq 0.05$, ***$p < 0.001$, ****$p < 0.0001$.

The online version of this article includes the following source data for figure 7:

**Source data 1.** Quantitation of total infected area per sample after infection of HFFs ectopically expressing RARRES3 or FLUC with CTG-GFP for 96 hr.

**Source data 2.** Quantitation of average number of infection foci formed after infection of HFFs ectopically expressing RARRES3 or FLUC with CTG-GFP for 96 hr.

**Source data 3.** Absorbance values for lactate dehydrogenase activity assays using supernatants from IFNγ pretreated WT or RARRES3$^{-/-}$ HFF cells infected with CTG-GFP.

**Source data 4.** Quantitation of total infected area per sample after infection of IFNγ pretreated WT or RARRES3$^{-/-}$ HFFs ectopically expressing RARRES3 or FLUC with CTG-GFP for 96 hr.

**Source data 5.** Absorbance values for lactate dehydrogenase activity assays using supernatants from IFNγ pretreated HFF cells infected with CTG-GFP in the presence of Compound 1.

Although our studies define a role for RARRES3 in promoting early egress in vitro, they do not address how this outcome impacts parasite infection in vivo. Premature egress prevents further parasite growth and would reduce the maximum number of parasites reduce per round of infection while also incurring additional energy costs of reinvasion. Further, it removes the parasite from its protected intracellular niche, resulting in exposure to the extracellular environment and potential recognition by the immune system. Finally, host cell death that accompanies premature egress is a proinflammatory event that would presumably lead to a heightened immune response (*Rock and Kono, 2008*). For these reasons, we expect that the early egress induced by RARRES3 would be protective against *T. gondii* infection in vivo. However, it is possible that rapid egress could result in faster spread to neighboring cells. It is also worth noting premature expulsion by phagocytic cells has been suggested to promote the dissemination of *T. gondii* and other pathogens such as *Cryptococcus neoformans* (*Seoane and May, 2020*; *Drewry and Sibley, 2019*). Further studies using in vivo models of infection are needed to clarify the role of RARRES3 in control of infection.

In addition to RARRES3, we found that ectopic expression of IRF1 was sufficient to restrict *T. gondii* infection. It was not surprising to us that we identified IRF1 in this screen considering the well-known role of IRF1 in the secondary induction of ISGs and other protective factors downstream of IFN signaling (*Feng et al., 2021*). Mice deficient in Irf1 were previously shown to be more susceptible to *T. gondii* infection (*Khan et al., 1996*). However, considering the limited number of ISGs found to restrict *T. gondii* infection in our screen, we were curious as to what genes or pathways were being induced by IRF1 ectopic expression and how this compared to IFNγ-mediated gene expression in A549s. Although few genes were substantially downregulated by IRF1 expression, 160 genes were upregulated ≥2-fold by IRF1 compared to 380 genes with IFNγ treatment. IRF1 induced genes were largely a subset of those IFNγ inducible genes. Notably, this is not always the case with a significant disparity in genes being induced by IRF1 compared to IFNs being reported previously in BEAS-2B cells (*Panda et al., 2019*). Overall, similar processes were induced by both IRF1 and IFNγ and these were related to antigen processing and presentation, host defense, and immune signaling. We observed that IRF1 and IFNγ both strongly induced the expression of RARRES3 which led us to question if infection restriction by IRF1 was due to the induction of RARRES3 expression. However, we found that this was not the case, suggesting that RARRES3 either does not play a role in IRF1-mediated restriction of *T. gondii* infection or plays a redundant role. Hence, it is likely that IRF1 leads to overexpression of multiple ISGs that collectively inhibit parasite growth.

Although our study is the first to broadly screen ISGs for anti-protozoal activity, it is curious that we only identified two genes that were inhibitory to *T. gondii* infection. In contrast, similar screens challenging viruses and bacteria have commonly identified a minimum of 2–3 times this number of gene products that were restrictive to infection (*Abrams et al., 2020*; *Schoggins et al., 2011*; *Schoggins et al., 2014*; *Perelman et al., 2016*; *Dittmann et al., 2015*; *Kuroda et al., 2020*; *Kane et al., 2016*; *Liu et al., 2019*). To our knowledge, a total of 30 viruses and two bacterial pathogens have been challenged in similar screens. It is possible that the above observation may not hold as more screens are performed on other pathogens especially intracellular bacteria or parasites. One possible explanation of this difference is that *T. gondii* separates itself from the host cytoplasm via the host derived PVM that forms a barrier to otherwise harmful ISGs in the cytosol. Another explanation is that a common mechanism of ISG activity is the manipulation or shutdown of host processes required for pathogen infection. Examples include PKR-mediated and IFIT-mediated inhibition of host translation machinery, processes that directly affect viral infection (*Schoggins, 2019*). In contrast, *T. gondii* is a relatively autonomous intracellular pathogen with relatively limited need for host machinery for its growth and replication. Hence, the fact that so few genes were identified by this screen may reflect a secluded existence of T. gondii within the PVM and its autonomy of cellular processes relative to viral or bacterial pathogens.

Alternatively, the low number of single genes that restrict the growth of *T. gondii* in A549 cells suggests that IFNγ-dependent restriction is a complex process requiring the cooperation of multiple host factors at a time. Hence, the expression of single factors may not be sufficient to restrict infection. For example, a requirement for additional factors could explain why ISG15 was not identified by this screen as ISGylation also requires ubiquitin-like conjugation to attach to target proteins (*Perng and Lenschow, 2018*). ISG15 knockout was previously found to enhance *T. gondii* growth in A549 cells (*Bhushan et al., 2020*) and this was attributed to its role in the targeting of autophagy machinery

to the PV during infection leading to restriction of parasite growth. Currently, over 200 genes have been found to be involved in autophagy in human cells according to the human autophagy database (HADb, http://autophagy.lu/clustering/index.html). Furthermore, coimmunoprecipitation experiments indicated that ISG15 interacts directly or indirectly with more than 240 proteins (*Bhushan et al., 2020*). Thus, although ISG15 may be necessary for autophagy-mediated infection restriction as observed with knockout-based studies, its individual ectopic expression may not be sufficient to induce this mechanism of infection restriction. A similar explanation can be made for GBP1 that was shown via knockout experiments to be important for restriction of *T. gondii* in A549 cells previously (*Johnston et al., 2016*). Gbp1 recruitment to the PV and parasite clearance in mice has been shown to require the autophagy machinery (*Selleck et al., 2013*) and the interaction of GBP1 with ATG5 has been reported in a human cell line (*Bhushan et al., 2020*). In human macrophages, the AIM2 inflammasome has been reported to induce host cell death in a GBP1 dependent manner (*Fisch et al., 2019a*). Thus, the expression of GBP1 individually may not be sufficient to restrict *T. gondii* infection. A similar argument is unlikely to explain the lack of a phenotype for IDO1, which is not thought to require any other genes for its function. However, this pathway does not seem to operate in all human cells (*Qin et al., 2017*; *Woodman et al., 1991*; *Dimier and Bout, 1997*). Our findings demonstrate that IDO1 does not impact infection in A549s at least under the conditions used in this study. Finally, it is also possible that ISGs important for *T. gondii* were missed by this screen as the library used herein is not an exhaustive list of all ISGs expressed in A549 cells. Hence, it is possible that other ISGs that restrict *T. gondii* infection could be identified using a more focused library or using a similar approach in a different human cell line.

Although we successfully identified a novel anti-parasitic ISG impacting *T. gondii* infection, the scarcity of ISGs identified with the screening approach used here is also of interest. Our results suggest that immunity to *T. gondii* is a complex process requiring multiple factors to impact infection. This complexity differs from other intracellular pathogens such as bacteria and viruses, where single ISGs are often sufficient to inhibit infection. As such, a future loss-of-function based screen for ISGs targeting *T. gondii* infection may reveal additional mechanisms of *T. gondii* restriction.

# Materials and methods

**Key resources table**

| Reagent type (species) or resource | Designation | Source or reference | Identifiers | Additional information |
|---|---|---|---|---|
| Cell line (*Homo sapiens*) | A549 | ATCC | CCL-185; RRID:CVCL_0023 | |
| Cell line (*H. sapiens*) | 293T | ATCC | CRL-3216; RRID:CVCL_0063 | |
| Cell line (*H. sapiens*) | HFF | ATCC | SCRC-1041; RRID:CVCL_3285 | |
| Cell line (*H. sapiens*) | HeLa | ATCC | CCL-2; RRID:CVCL_0030 | |
| Genetic reagent (*H. sapiens*) | HeLa reporter cell line expressing ISRE-GLUC | PMID:31413201 | 11×-ISRE-Gaussia Luciferase reporter line | |
| Genetic reagent (*H. sapiens*) | HeLa reporter cell line expressing GAS-FLUC | PMID:27091930 | HeLa reporter cell line expressing GAS-FLUC | |
| Genetic reagent (*H. sapiens*) | RARRES3$^{-/-}$ Lung adenocarcinoma (A549) | This paper | | See Materials and methods |
| Genetic reagent (*H. sapiens*) | IDO1$^{-/-}$ Lung adenocarcinoma (A549) | This paper | | See Materials and methods |
| Strain, strain background (*Toxoplasma gondii*) | RH88 | ATCC | 50853 | |
| Strain, strain background (*T. gondii*) | Me49 | ATCC | 50611 | |
| Strain, strain background (*T. gondii*) | CTG | ATCC | 50842 | |
| Genetic reagent (*T. gondii*) | RH88-GFP | This paper | | See Materials and methods |

*Continued on next page*

*Continued*

| Reagent type (species) or resource | Designation | Source or reference | Identifiers | Additional information |
|---|---|---|---|---|
| Genetic reagent (*T. gondii*) | Me49-GFP | This paper | | See Materials and methods |
| Genetic reagent (*T. gondii*) | CTG-GFP | This paper | | See Materials and methods |
| Antibody | Anti-RFP (rabbit polyclonal) | Thermo Fisher Scientific | Cat#: R10367; RRID:AB_2315269 | (1:2000) |
| Antibody | Anti-GFP (mouse monoclonal) | Thermo Fisher Scientific | Cat#: A-11120; RRID:AB_221568 | (1:2000) |
| Antibody | Anti-mouse Alexa Fluor 488 (goat polyclonal) | Life Technologies | Cat#: A-11029; RRID:AB_138404 | (1:1000) |
| Antibody | Anti-rabbit Alexa Fluor 568 (goat polyclonal) | Life Technologies | Cat#: A-11011; RRID:AB_143157 | (1:1000) |
| Antibody | Anti-rabbit Alexa Fluor 488 (goat polyclonal) | Invitrogen | Cat#: A-11034; RRID:AB_2576217 | (1:1000) |
| Antibody | Anti-V5 (mouse monoclonal) | Invitrogen | Cat#: R960-25; AB_2556564 | (1:5000) |
| Antibody | Anti-mouse 680RD (goat polyclonal) | LI-COR Biotechnology | Cat#: 926-68070; RRID:AB_10956588 | (1:5000) |
| Antibody | Anti-rabbit 800CW (goat polyclonal) | LI-COR Biotechnology | Cat#: 926-32211; RRID:AB_621843 | (1:5000) |
| Antibody | Anti-actin (mouse monoclonal) | Sigma-Aldrich | Cat#: MAB1501; RRID:AB_2223041 | (1:5000) |
| Antibody | Anti-IDO1 (rabbit monoclonal) | Cell Signaling Technology | Cat#: 86630S; RRID:AB_2636818 | (1:1000) |
| Antibody | Anti-IRF1 D5E4 (rabbit monoclonal) | Cell Signaling Technology | Cat#: 8478S; RRID:AB_10949108 | (1:500) |
| Chemical compound, drug | Necrosulfonamide; NSA | Tocris Biotechnology | Cat#: 5025 | (10 µM) |
| Chemical compound, drug | GSK'872 | Selleck Chemicals | Cat#: S8465 | (5 µM) |
| Chemical compound, drug | Z-YVAD-FMK | Sigma-Aldrich | Cat#: 218,746 | (10 µM) |
| Chemical compound, drug | Z-VAD-FMK | R&D Systems | Cat#: fmk001 | (50 µM) |
| Chemical compound, drug | GSK'963 | Selleck Chemicals | Cat#: S8642 | (1 µM) |
| Chemical compound, drug | Compound 1 | MedChemExpress | Cat#: HY-101525 | (5 µM) |
| Chemical compound, drug | Hoechst stain; Hoechst 33342 | Invitrogen | Cat#: H3570 | (1 µg/ml) |
| Chemical compound, drug | SYTOX Green | Invitrogen | Cat#: S7020 | (5 µM) |
| Chemical compound, drug | XtremeGene9 DNA Transfection Reagent | Roche | Cat#: XTG9-RO | |
| Recombinant DNA reagent | pHAGE NFkB-TA-LUC-UBC-GFP-W (plasmid) | Addgene | 49,343 | |
| Recombinant DNA reagent | plentiCRISPRv2 (plasmid) | Addgene | 52,961 | |
| Recombinant DNA reagent | pLenti-Cas9-GFP (plasmid) | Addgene | 86,145 | |

*Continued on next page*

*Continued*

| Reagent type (species) or resource | Designation | Source or reference | Identifiers | Additional information |
|---|---|---|---|---|
| Recombinant DNA reagent | pTRIP.RARRES3 (plasmid) | PMID:21478870 | pTRIP.CMV.IVSb.RARRES3.ires.TagRFP | |
| Recombinant DNA reagent | pTRIP.V5-RARRES3 WT (plasmid) | This paper | | See Materials and methods |
| Recombinant DNA reagent | pTRIP.V5-RARRES3 C113A (plasmid) | This paper | | See Materials and methods |
| Recombinant DNA reagent | pTRIP.V5-RARRES3 C113S (plasmid) | This paper | | See Materials and methods |
| Recombinant DNA reagent | pDONR221 (plasmid) | Invitrogen | Cat#: 12536017 | |
| Recombinant DNA reagent | pTRIP.FLUC (plasmid) | PMID:21478870 | pTRIP.CMV.IVSb.FLUC.ires.TagRFP | |
| Recombinant DNA reagent | pTRIP.IRF1 (plasmid) | PMID:21478870 | pTRIP.CMV.IVSb.IRF1.ires.TagRFP | |
| Recombinant DNA reagent | pTRIP.GBP2 (plasmid) | PMID:21478870 | pTRIP.CMV.IVSb.GBP2.ires.TagRFP | |
| Recombinant DNA reagent | pVSVg (plasmid) | PMID:21478870 | Plasmid expressing the vesicular stomatitis virus glycoprotein (VSVg) | |
| Recombinant DNA reagent | pGag-pol (plasmid) | PMID:21478870 | Plasmid expressing HIV gag-pol | |
| Recombinant DNA reagent | pGRA1.GFP.GRA2.DHFR (plasmid) | This paper | | See Materials and methods |
| Commercial assay or kit | CyQuant LDH Cytotoxicity Assay Kit | Invitrogen | Cat#: C20300 | |
| Commercial assay or kit | Luciferase cell culture lysis reagent (5×) | Promega | Cat#: E1531 | |
| Commercial assay or kit | Luciferase Assay System | Promega | Cat#: E1500 | For firefly luciferase |
| Commercial assay or kit | Pierce Gaussia Luciferase Glow Assay Kit | Thermo Fisher Scientific | Cat#: 16160 | For Gaussia luciferase |
| Peptide, recombinant protein | Q5 High Fidelity DNA Polymerase | NEB | Cat#: M0491 | Recombinant fragment and plasmid amplification |
| Peptide, recombinant protein | PrimeSTAR GXL DNA Polymerase | Tocris Biotechnology | Cat#: R050 | Genomic DNA amplification |
| Peptide, recombinant protein | BsmBI-v2 | NEB | Cat#: R0739 | |
| Peptide, recombinant protein | BP Clonase II | Thermo Fisher Scientific | Cat#: 11789-020 | |
| Peptide, recombinant protein | LR Clonase II | Thermo Fisher Scientific | Cat#: 11791-020 | |
| Software, algorithm | CellProfiler | BROAD Institute | RRID:SCR_007358 | Version 3.1.9 |
| Software, algorithm | GraphPad Prism | GraphPad | RRID:SCR_002798 | Version 3.1.2 |
| Software, algorithm | ImageJ | National Institutes of Health | RRID:SCR_003070 | Version 1.53f51 |
| Software, algorithm | ZEN Blue | Zeiss | RRID:SCR_013672 | Version 2.5 |

## Cell lines and parasites

HeLa adenocarcinoma, A549 lung carcinoma (ATCC #CCL-185), HFF foreskin fibroblast (ATCC #SCRC-1041), and human embryonic kidney-derived 293T cells (ATCC #CRL-3216) were grown in Dulbecco's

modified Eagle's medium (DMEM) supplemented with 10% fetal bovine serum (FBS), 10 mM HEPES (pH 7.5), 2 mM L-glutamine, and 10 µg/ml gentamicin. Cells were grown at 37°C with 5% $CO_2$. *T. gondii* strains RH88 (type I), Me49 (type II), and CTG (type III) expressing GFP were generated via random insertion of pGRA1.GFP.GRA2.DHFR after electroporation as described previously (*Shen et al., 2017*). Clonal populations expressing GFP were generated via limiting dilution. *T. gondii* lines were passaged as described previously in HFFs grown under the conditions listed above (*Khan and Grigg, 2017*). Parasite and host cell lines were confirmed to be negative for mycoplasma using an e-Myco Plus Kit (Intron Biotechnology).

## Plasmids and cloning

The plasmids TRIP.RARRES3 and control constructs were kindly provided by Neal Alto and John Schoggins. Briefly, the TRIP plasmid encodes an expression cassette flanked by lentiviral LTRs. Expression of a bicistronic transcript including tagRFP and a gene of interest is driven by a CMV promoter. The gene of interest and tagRFP are translated independently via an internal ribosome entry site. Cas9 resistant RARRES3 was generated from the WT TRIP.RARRES3 construct by overlap extension PCR using Q5 high fidelity DNA polymerase (NEB). N-terminally V5 tagged WT, C113A, and C113S RARRES3 were generated from WT by overlap extension PCR. Fragments were cloned into pDONR221 using BP Clonase II (Thermo Fisher Scientific) and subsequently cloned into pTRIP using LR Clonase II (Thermo Fisher Scientific) according to the manufacturer's protocol. For CRISPR/Cas9 experiments, RARRES3 and nontargeting guides were cloned into plentiCRISPRv2 (Addgene plasmid #52961) (*Sanjana et al., 2014*). Primers for the above cloning are listed in *Supplementary file 4*. For IDO1, an IDO1 targeting sgRNA was cloned into pLenti-Cas9-GFP (Addgene plasmid #86145). Briefly, pLenti-Cas9-GFP was digested with BsmBI (New England Biolabs). Primers listed in *Supplementary file 4* were annealed and ligated into the digested plasmid using T4 ligase (New England Biolabs). For the generation of pGRA1.GFP.GRA2.DHFR, an expression cassette consisting of the GRA1 5′ UTR (M26007.1, nucleotides 4–615) driving the expression of GFP (MN114103.1, coding sequence) flanked by the GRA2 3′ UTR (XM_002366354.2, nucleotides 997–1114) was cloned into pHL931 along with DHFR (XM_002367211.2, coding sequence) expressed from its native promoter and flanked by its 3′ UTR (L08489.1).

## Lentivirus production and cell line generation

TRIP lentiviruses were produced as previously described (*Schoggins et al., 2012*). Lentiviruses derived from Lenti-Cas9-GFP, lentiCRISPRv2, and HAGE NFkB-TA-LUC-UBC-GFP-W (*Wilson et al., 2013*) were produced similarly. Briefly, 293T cells were seeded at $4×10^5$ cells per well into six-well plates. Cells were transfected with 1 µg pTRIP, pLenti-Cas9-GFP, plentiCRISPRv2, or pHAGE NFkB-TA-LUC-UBC-GFP-W, 0.2 µg plasmid expressing VSVg, and 0.8 µg plasmid expressing HIV-1 gag-pol using X-tremeGENE 9 (Sigma-Aldrich). Media were changed 6 hr later and lentivirus containing culture supernatants were collected at 48 and 72 hr post-transfection. Pooled supernatants were clarified by centrifugation at 800×*g* for 5 min. Polybrene and HEPES were added to a final concentration of 4 µg/ml and 35 mM, respectively. Lentivirus was stored at –80°C until use.

For lentivirus transductions, cells were seeded at $7×10^4$ cells per well in 24-well plates. The next day, media were changed to DMEM supplemented with 4 µg/ml polybrene, 3% FBS, 35 mM HEPES, 2 mM glutamine, and 10 µg/ml gentamicin. Cells were transduced by spinoculation at 800×*g*, 45 min, 37°C. For the ISG screen, media was changed 6 hr later to normal growth medium. Cells were replated at 48 hr post-transduction for subsequent experimentation.

For knockout cell line generation, cells were transduced with lentiCRISPRv2 or pLenti-Cas9-GFP containing the appropriate sgRNA for Cas9 targeting as above. For lentiCRISPRv2, cells were selected for at least 2 weeks in growth media containing 4 µg/ml puromycin before experimentation. For IDO1⁻ᐟ⁻, STAT1⁻ᐟ⁻, and RARRES3⁻ᐟ⁻ A549 cell lines, cells were transduced with a single lentivirus and clonal cell lineages were established through limiting dilution. For HFFs, a heterogeneous bulk population RARRES3 knockout cell line was generated by transducing at a tissue culture infectious dose of 90% ($TCID_{90}$) with two different lentiCRISPRv2 based lentiviruses expressing separate RARRES3 targeting sgRNAs. Nontargeting control cell lines were generated for use as a control in all experiments involving RARRES3⁻ᐟ⁻ cells. Here, cells were transduced with a single lentiCRISPRv2 based lentivirus containing a single nontargeting guide. For RARRES3 and STAT1, editing was confirmed

by PCR amplifying targeted loci using primers listed in *Supplementary file 4* and PrimeSTAR GXL DNA polymerase (Tocris) followed by Sanger sequencing. Editing efficiency was quantitated using Synthego ICE analysis (https://ice.synthego.com/#/). For single-cell clones, >90% editing was verified. For HFF bulk population knockout of RARRES3, 68% editing of sequenced alleles was observed. For STAT1, editing was further confirmed functionally via testing the sensitivity of cells to IFN treatment as determined by IRF1 induction. For IDO1, editing was confirmed via loss of protein expression observed by western blot.

## Infections

A549 and HeLa cells were seeded at $1.5 \times 10^4$ in 96-well plates 24 hr prior to infection. HFFs were seeded at $2 \times 10^4$ in 96-well plates 24 hr prior to infection. Cells were infected with parasites diluted in 200 μl normal growth medium for 1 hr at 37°C. Medium was subsequently changed to 300 μl normal growth medium. For single life cycle infections (typically indicated as 36 hr infections), an MOI of 1 was used. For focus forming assays (typically indicated as 96 hr infections), an MOI of 0.03 was used. For LDH assays, media were changed to 200 μl normal growth medium. For experiments involving IFNγ, cells were pretreated with or without IFNγ diluted in normal growth medium as indicated for 24 hr prior to infection. For infections involving cell death inhibitors or Compound 1, drugs were added during the media change after the 1-hr infection period. For imaging-based experiments, cells were fixed in 4% formaldehyde for 10 min after infection and washed with phosphate-buffered saline (PBS) before subsequent experimentation.

## Drugs

Stocks of the cell death inhibitors Z-VAD-FMK (R&D Systems), GSK'963 (Selleck Chemicals), GSK'872 (Selleck Chemicals), NSA (Tocris), and Z-YVAD-FMK (Sigma-Aldrich) as well as Compound 1 (obtained from MERCK & CO., Inc) were prepared in DMSO. For use, the drugs were diluted in normal growth medium to the following working concentrations: Z-VAD-FMK (50 μM), GSK'963 (1 μM), GSK'872 (5 μM), NSA (10 μM), Z-YVAD-FMK (10 μM), and Compound 1 (5 μM). A DMSO control was included in experiments involving these drugs with a final DMSO concentration of 1%.

## LDH assays

LDH assays were performed with the CyQuant LDH Cytotoxicity Assay Kit (Invitrogen) according to the manufacturer's protocol. Briefly, A549 or HFF cells split in 96-well plates were infected for 1 hr as described above with CTG-GFP at an MOI of 40 or 15, respectively, and subsequently treated with drugs as indicated. After 36 or 72 hr, 20 μl of 10× lysis buffer or PBS was added to each well and incubated at 37°C for 30 min. Afterward, 50 μl of cell supernatant was mixed with 50 μl of assay buffer and substrate for 30 min at room temperature. The reaction was stopped with 50 μl stop solution and absorbance was measured at 490 nm.

## Luciferase assays

Previously generated HeLa cells expressing GAS-FLUC, GFP-FLUC, or ISRE-GLUC reporters were transduced with TRIP.RARRES3 or TRIP.FLUC lentivirus as described above (*Bando et al., 2018*; *Olias and Sibley, 2016*). For kB-LUC, HeLa cells were additionally transduced with HAGE NFkB-TA-LUC-UBC-GFP-W lentivirus. After 48 hr, cells were split into 96-well plates at $1.5 \times 10^4$ cells/well. Cells were treated with or without 100 U/ml IFNβ or IFNγ as indicated for 24 hr and subsequently infected as indicated with CTG-GFP at an MOI of 2 for 24 hr. For firefly luciferase assays, cells were lysed in 50 μl of 1× Luciferase Cell Culture Lysis Buffer (Promega). For Gaussia luciferase assays, supernatant was collected. Luciferase assays were conducted using Pierce Gaussia Luciferase Glow Assay Kit (Thermo Fisher Scientific) or Luciferase Assay System Kit (Promega) according to the manufacturer's protocol.

## Next generation RNA-sequencing sample preparation and analysis

A549 cells transduced with TRIP.IRF1 or TRIP.FLUC derived lentivirus were split at $3.5 \times 10^6$ into 100 mm dishes. After 24 hr, cells were treated with or without IFNγ at 1000 U/ml for an additional 12 hr before harvest with RLT buffer. RNA was isolated with a Qiagen RNeasy Mini Kit according to the manufacturer's protocol. Prior to sequencing, RNA quality was determined on an Agilent Bioanalyzer to have a RIN>8.0. Libraries prepared from samples were analyzed with an Illumina NovaSeq 6000 S4

generating a minimum of $3\times10^7$ reads per sample. Data were analyzed with Partek Flow software. Prior to alignment, 5 bp were trimmed from the 5′ end of transcripts. Only fragments ≥25 bp in length were considered for alignment. Alignment was conducted with the STAR aligner and differential expression analysis was conducted using GSA analysis with recommended settings. Genes were characterized here as induced by IFNγ or IRF1 if they induced gene expression ≥2-fold with an FDR<0.05. Gene lists were compared using GeneVenn (http://genevenn.sourceforge.net/index.htm; *Pirooznia et al., 2007*). For GO analysis, gene lists were analyzed with the PANTHER classification system using the GO biological process complete data set (*Thomas et al., 2003*; *Thomas et al., 2006*). Statistical significance was determined with Fisher's exact test using the Bonferroni correction for multiple testing. Only processes containing at least 25 total genes with a p-value≤0.05 were considered.

## Immunofluorescence and imaging

Samples were fixed in 4% formaldehyde for 10 min at room temperature. Wash buffer (WB) consisted of 1% FBS, 1% normal goat serum (NGS), and 0.02% Saponin in PBS. Samples were blocked for 30 min with PBS containing 5% FBS, 5% NGS, and 0.02% Saponin. Samples were incubated with 1:2000 anti-RFP antibody (Invitrogen) and 1:2000 anti-GFP (Invitrogen) in WB overnight, washed four times in WB for 5 min each, and probed with 1:1000 goat anti-mouse Alexa Fluor 488 (Life Technologies) and 1:1000 goat anti-rabbit Alexa Fluor 568 (Life Technologies) in WB for 1 hr. For IRF1 staining, 1:500 anti-IRF1 primary antibody (Cell Signaling Technology) and 1:1000 anti-rabbit Alexa Fluor 488 secondary antibody (Life Technologies) were used instead. Samples were washed three times with WB and nuclei were stained for 5 min with Hoechst 33342 (Life Technologies) in WB. Samples were imaged with a Cytation 3 Imager (BioTek) and images were analyzed in CellProfiler v3.1.9.

For live imaging experiments, cells were imaged starting 2 hr postinfection every 15 min for 48 hr at 37°C under 5% $CO_2$. Live imaging was conducted with a Zeiss Observer Z1 inverted microscope (Zeiss) using a Colibri 7 LED light source (Zeiss), ORCA-ER digital camera (Hamamatsu Photonics), Plan-Neofluar 10× (NA 0.3) objective (Zeiss), and ZEN Blue image acquisition software (v2.5). Following acquisition, channel bleed-through was subtracted from GFP channel images and images were contrast enhanced using ImageJ (v1.53f51).

## Western blotting

A549 cells were split at $2.5\times10^5$ cells per well into six-well plates in standard growth medium. The following day, media were changed to standard growth medium supplemented with or without 1000 U/ml IFNγ. After 24 hr, cells were washed with PBS, trypsinized with 0.05% trypsin, and spun down at 200×*g* for 5 min. Cell pellets were washed once with PBS and lysed with CelLytic M (Millipore) supplemented with 20 mM DTT and 125 U/ml benzonase (Millipore). Samples were incubated at room temperature for 20 min, run on a 10%, 37.5:1 polyacrylamide gel, and transferred to nitrocellulose membranes. Membranes were blocked with 0.1% Tween-20 PBS-T containing 5% bovine serum albumin (BSA) for 30 min. Membranes were incubated with 1:1000 rabbit anti-IDO1 (Cell Signaling Technology), 1:5000 mouse anti-V5 (Invitrogen), or 1:5000 mouse anti-actin (Sigma-Aldrich) in 5% BSA PBS-T for 1 hr, washed four times with PBS-T for 3 min each, incubated with 1:5000 goat anti-mouse 680RD (LI-COR) and 1:5000 goat anti-rabbit 800CW (LI-COR) in 5% BSA PBS-T for 30 min, washed four times in PBS-T for 3 min each, and washed two times in PBS. Membranes were imaged with a LI-COR Odyssey scanner.

## Statistical analysis

For most data sets including those normalized to control, statistical significance was determined with a two-way ANOVA and Tukey's honestly significant difference post hoc test conducted on raw data prior to normalization and considered variance between experimental replicates and variance between experimental conditions. For LDH and luciferase reporter experiments, statistical significance was determined after normalization and considered variance between experimental replicates and variance between experimental conditions. For experiments in *Figure 2A and B*, statistical significance was determined using a Brown-Forsythe and Welch ANOVA. Specifically for data sets not normalized to control with only two conditions, a Mann-Whitney U-test was used to determine statistical significance. All statistical analyses were conducted using GraphPad Prism (v3.1.2). The term 'technical replicate' refers to separate samples derived from the same original source within the same experiment

(i.e., wells of a plate) processed on the same day. The term 'biological replicate' refers to separate experiments conducted on different dates with different samples.

## Acknowledgements

The authors thank members of the Sibley lab for helpful suggestions. Partially supported by NIH grants (R21 AI154048, R01 AI118426 to LDS). The Welch Foundation (I-1704 to NMA) and the National Institutes of Health (AI083359 to NMA).

---

## Additional information

### Competing interests

John W Schoggins: Reviewing editor, *eLife*. The other authors declare that no competing interests exist.

### Funding

| Funder | Grant reference number | Author |
|---|---|---|
| National Institutes of Health | AI154048 | L David Sibley |
| National Institutes of Health | AI118426 | L David Sibley |
| National Institutes of Health | AI083359 | Neal M Alto |
| Welch Foundation | I-1704 | Neal M Alto |
| National Institutes of Health | AI158124 | John W Schoggins |

The funders had no role in study design, data collection and interpretation, or the decision to submit the work for publication.

### Author contributions

Nicholas Rinkenberger, Conceptualization, Formal analysis, Investigation, Methodology, Validation, Visualization, Writing - original draft, Writing - review and editing; Michael E Abrams, Methodology, Resources; Sumit K Matta, John W Schoggins, Neal M Alto, Methodology, Resources, Writing - review and editing; L David Sibley, Conceptualization, Funding acquisition, Project administration, Supervision, Writing - review and editing

### Author ORCIDs

John W Schoggins http://orcid.org/0000-0002-7944-6800
Neal M Alto http://orcid.org/0000-0002-7602-3853
L David Sibley http://orcid.org/0000-0001-7110-0285

### Decision letter and Author response

Decision letter https://doi.org/10.7554/eLife.73137.sa1
Author response https://doi.org/10.7554/eLife.73137.sa2

---

## Additional files

### Supplementary files

• Supplementary file 1. Summary of overexpression screen in A549 cells.

• Supplementary file 2. List of genes induced by IRF1 ectopic expression or IFNγ treatment in A549 cells.

• Supplementary file 3. Comparison of genes induced by IRF1 or IFNγ with genes in the type II ISG screen library.

- Supplementary file 4. List of primer sets used.
- Transparent reporting form
- Source data 1. GFP channel of live video microscopy for CTG-GFP infection in A549 cells ectopically expressing FLUC control.
- Source data 2. Phase channel of live video microscopy for CTG-GFP infection in A549 cells ectopically expressing FLUC control.
- Source data 3. GFP channel of live video microscopy for CTG-GFP infection in A549 cells ectopically expressing RARRES3 control.
- Source data 4. Phase channel of live video microscopy for CTG-GFP infection in A549 cells ectopically expressing RARRES3 control.

### Data availability

RNASeq data generated here have been deposited to GEO with the accession number GSE181861.

The following dataset was generated:

| Author(s) | Year | Dataset title | Dataset URL | Database and Identifier |
|---|---|---|---|---|
| Sibley LD | 2021 | Over-expression screen of interferon-stimulated genes identifies RARRES3 as a restrictor of Toxoplasma gondii infection | http://www.ncbi.nlm.nih.gov/geo/query/acc.cgi?acc=GSE181861 | NCBI Gene Expression Omnibus, GSE181861 |

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
