## [Editor Report]

*Toxoplasma gondii* is a widespread parasite of warm-blooded animals, with estimates suggesting 2 billion people are currently and chronically infected with this pathogen. Many questions remain as to how humans control and eliminate *T. gondii* following infection. In this manuscript, Rinkenberger et al. reveal a previously unidentified and understudied host factor, RARRES3 that promotes cell autonomous control of *T. gondii* in human cells by mediating pathogen expulsion from infected cells.

---

## [Decision Letter]

**Decision letter after peer review:**

Thank you for submitting your article "Over-expression Screen of Interferon-Stimulated Genes Identifies RARRES3 as a Restrictor of *Toxoplasma gondii* Infection" for consideration by *eLife*. Your article has been reviewed by 2 peer reviewers, and the evaluation has been overseen by a Reviewing Editor and Jos Van der Meer as the Senior Editor. The following individual involved in review of your submission has agreed to reveal their identity: Kirk Jensen (Reviewer #2).

The reviewers have discussed their reviews with one another, and the Reviewing Editor has drafted this to help you prepare a revised submission. Overall there is considerable enthusiasm for the quality and importance of the work and we would be delighted to entertain a revised version if the following essential revisions can be addressed.

Essential revisions:

1) There are no data in the paper that address the importance of the observations in the context of an in vivo infection. We have decided that such experiments are not required for a revised version. However, in the absence of such data, it is impossible to know whether the observed process is anti-parasite or perhaps beneficial to the parasite. Although pathogen expulsion is most likely anti-pathogen, it has also been the proposed that process of pathogen expulsion may benefit the pathogen (see microreview by Seoane and May PMID: 31730731). Tempering of the claims of an anti-parasite function of RARRES3 is thus required as is an acknowledgement that the effects of RARRES3 during in vivo infection remain unknown. These points could be discussed in the revised manuscript.

2) In the absence of in vivo data, the most novel aspect of the paper would be a demonstration that RARRES3 is involved in pathogen expulsion. However, the data in the manuscript in support of this mechanism are all indirect. Some direct observational data demonstrating that RARRES3 acts to mediate explusion (live imaging perhaps?) seem to us as an essential requirement in a revised manuscript.

3) The invidividual reviewers identify additional changes to the manuscript that should also be addressed in a revised version. Experiments with an additional pathogen would broaden the impact of the work but would not be required in a revised version.

*Reviewer #1 (Recommendations for the authors):*

Citations:

– Lines 55/56: would be helpful to cite one or more review articles on IRGs/ GBPs.

– Line 57: instead of citing 2 review articles, I would suggest the authors cite the 3 original papers that first demonstrated the Toxo PV-oriented membrano-lytic activity of IRGs/ GBPs: PMID: 16940170; PMID: 16304607; PMID: 22795875; after all the manuscript cites the original Toxoplasma work on IFNγ, nitric oxide, IDO, etc.- so why not do the same for IRGs/ GBPs?

– Line 58: why cite a 2012 review considering that many relevant ROP18-IRG papers have been published since 2012? I'd recommend PMID: 33178630 for a more timely review. Alternatively, cite the most relevant original research papers.

Data presentation:

– It is most common to have the controls (untreated, DMSO, WT, FLUC, etc.) on the left. However, it's also fine to have the controls on the right, if that's your preference, but whatever the arrangement, it should be consistent across all panels / figures to make it easier/ more intuitive to read the figures. For examples: Figure 5 – going from left to right – KO first and then WT in panels A and B but WT first and then KO in panel C. In Figure 1 panel A data are shown as IFN treated on the left and untreated on the right but in panel F it is the other way around. Etc. Please, check all figures and make it consistent across all figures and panels.

Writing/ argumentation:

– Line 167: PKG is mentioned but not defined. Definition is provided in next sentence.

– Lines 177/178: I am having a hard time understanding what the authors are trying to argue here. Admittedly, that could be me. Consider editing?

– First sentence of the Discussion: "IFNγ-mediated immunity are highly conserved across different cell types in mice," – does this statement specifically refer to immunity to Toxo? Also, what's the evidence for this to be true? Provide citations.

– When comparing mouse with humans, the authors should also consider that most experiments in mice have been performed in primary cells (astrocytes, fibroblasts), whereas most experiments in human cells have been conducted in immortalized cancer cells – often of epithelial origin (e.g. A549 in this study) – therefore, these broad statements regarding host species-specific conservation of immune responses (or lack thereof) seem to be based on an incommensurable comparison

– Last sentence of first paragraph of Discussion "Hence the control of intracellular eukaryotic pathogens contrasts with that of bacterial and viral pathogens, mirroring differences in their overall biological complexity" – What is the evidence that the control of intracellular eukaryotic pathogens is fundamentally different from the control of intracellular bacterial pathogens – let's say Chlamydia, for example. So far observations made with Toxo closely mirror what has been seen with Chlamydia, a bacterium, and vice versa. I would suggest to revise this statement.

– In the same vain: line 253 ff. "In contrast, similar screens challenging viruses and bacteria have commonly identified a minimum of 2-3 times this number of ISGs that were restrictive to infection(10-13). […]" The number of screens is small and doesn't justify drawing such sweeping conclusions. If I am not mistaken, there's only been one screen with a bacterial pathogen, *Listeria*. I would suggest the authors rethink this entire paragraph (and other text passages such as the final paragraph of the Discussion), as I suspect time will show that these broad statements are inaccurate.

– Could the authors provide a rationale for choosing a type III strain for their screen

– Discuss potential physiological relevance of the egress pathway. Could this actually benefit the parasite in vivo by escaping the toxic intracellular environment of an IFNγ-primed cells? Would ejected parasites be more easily killed by phagocytes?

Suggested experiments/ data analysis

– REF [57] showed timelapse microscopy capturing parasite egress in IFNγ primed cells. Similar experiments monitoring egress in real time in controls vs RARRES3 overexpression and/or WT vs RARRES3 KO cells would provide direct evidence in support of the proposed mechanistic model and therefore significantly strengthen the paper. Current evidence is entirely circumstantial.

– Host-driven/ host-protective egress pathways have been reported for other infectious agents (cryptococcus ['vomocytosis'], UEPC,.…) – it would broaden the impact of the study, if the authors were to test whether RARRES3 is also involved in the egress of one or two of these pathogens. Regardless of the outcome of these studies, the results would be highly informative. Authors have done these types of comparisons in previous follow-up studies to similar overexpression screens (see *Listeria* ISG overexpression screen and follow-up with Shigella).

– Provide statistical comparison of complemented KOs vs controls (KO + FLUC) under IFNγ primed conditions (Figure 5B). If study is underpowered, conduct more repeats. If no complementation is achieved (is achievable), then explain why.

---

## [Author Response]

Essential revisions:1) There are no data in the paper that address the importance of the observations in the context of an in vivo infection. We have decided that such experiments are not required for a revised version. However, in the absence of such data, it is impossible to know whether the observed process is anti-parasite or perhaps beneficial to the parasite. Although pathogen expulsion is most likely anti-pathogen, it has also been the proposed that process of pathogen expulsion may benefit the pathogen (see micro-review by Seoane and May PMID: 31730731). Tempering of the claims of an anti-parasite function of RARRES3 is thus required as is an acknowledgement that the effects of RARRES3 during in vivo infection remain unknown. These points could be discussed in the revised manuscript.

Extending these studies to test the role or RARRES3 in vivo is not straightforward, especially since control mechanism vary between humans and experimental models like mice. However, we agree that our findings that RARRES3 controls early egress, and apparent growth restriction in vitro, as necessarily indicative of control in vivo. We have added a discussion of the different potential roles for early egress – or expulsion, on either control of infection or potential spread of the parasite.

2) In the absence of in vivo data, the most novel aspect of the paper would be a demonstration that RARRES3 is involved in pathogen expulsion. However, the data in the manuscript in support of this mechanism are all indirect. Some direct observational data demonstrating that RARRES3 acts to mediate explusion (live imaging perhaps?) seem to us as an essential requirement in a revised manuscript.

We agree that this would be test of our assumption and we have performed a time-lapse study that demonstrates early egress in RARRES3 over-expression cells. The data are now included in figure 6C and videos 1-2.

3) The invidividual reviewers identify additional changes to the manuscript that should also be addressed in a revised version. Experiments with an additional pathogen would broaden the impact of the work but would not be required in a revised version.

We have addressed the remaining concerns with revisions to the text as detailed in the point by point response below.

Reviewer #1 (Recommendations for the authors):Citations:– Lines 55/56: would be helpful to cite one or more review articles on IRGs/ GBPs.

Review articles cited on line 57 were moved to line 56.

– Line 57: instead of citing 2 review articles, I would suggest the authors cite the 3 original papers that first demonstrated the Toxo PV-oriented membrano-lytic activity of IRGs/ GBPs: PMID: 16940170; PMID: 16304607; PMID: 22795875; after all the manuscript cites the original Toxoplasma work on IFNγ, nitric oxide, IDO, etc. – so why not do the same for IRGs/ GBPs?

We have added the suggested citations.

– Line 58: why cite a 2012 review considering that many relevant ROP18-IRG papers have been published since 2012? I'd recommend PMID: 33178630 for a more timely review. Alternatively, cite the most relevant original research papers.

We have added the suggested citation.

Data presentation:– It is most common to have the controls (untreated, DMSO, WT, FLUC, etc.) on the left. However, it's also fine to have the controls on the right, if that's your preference, but whatever the arrangement, it should be consistent across all panels / figures to make it easier/ more intuitive to read the figures. For examples: Figure 5 – going from left to right – KO first and then WT in panels A and B but WT first and then KO in panel C. In Figure 1 panel A data are shown as IFN treated on the left and untreated on the right but in panel F it is the other way around. Etc. Please, check all figures and make it consistent across all figures and panels.

Controls are now consistently placed on the left in all figures.

Writing/ argumentation:– Line 167: PKG is mentioned but not defined. Definition is provided in next sentence.

The suggested correction has been made.

– Lines 177/178: I am having a hard time understanding what the authors are trying to argue here. Admittedly, that could be me. Consider editing?

Additional explanation has been added to the text.

– First sentence of the Discussion: "IFNγ-mediated immunity are highly conserved across different cell types in mice," – does this statement specifically refer to immunity to Toxo? Also, what's the evidence for this to be true? Provide citations.

The statement has been toned down to better reflect the cell types tested thus far and additional citations were added.

– When comparing mouse with humans, the authors should also consider that most experiments in mice have been performed in primary cells (astrocytes, fibroblasts), whereas most experiments in human cells have been conducted in immortalized cancer cells -often of epithelial origin (e.g. A549 in this study) – therefore, these broad statements regarding host species-specific conservation of immune responses (or lack thereof) seem to be based on an incommensurable comparison

We added a statement in the Introduction noting this as an additional factor that complicates the comparison of results in mice and humans.

– Last sentence of first paragraph of Discussion "Hence the control of intracellular eukaryotic pathogens contrasts with that of bacterial and viral pathogens, mirroring differences in their overall biological complexity" – What is the evidence that the control of intracellular eukaryotic pathogens is fundamentally different from the control of intracellular bacterial pathogens – let's say Chlamydia, for example. So far observations made with Toxo closely mirror what has been seen with Chlamydia, a bacterium, and vice versa. I would suggest to revise this statement.

The statement has been revised as suggested.

– In the same vain: line 253 ff. "In contrast, similar screens challenging viruses and bacteria have commonly identified a minimum of 2-3 times this number of ISGs that were restrictive to infection(10-13). […]" The number of screens is small and doesn't justify drawing such sweeping conclusions. If I am not mistaken, there's only been one screen with a bacterial pathogen, Listeria. I would suggest the authors rethink this entire paragraph (and other text passages such as the final paragraph of the Discussion), as I suspect time will show that these broad statements are inaccurate.

We agree that the number of bacterial pathogens tested is very limited. We have clarified in the text the number of viral and bacterial pathogens which have been tested and noted that further work is needed to validate our observation.

– Could the authors provide a rationale for choosing a type III strain for their screen

A rationale has been added to the beginning of the Results section.

– Discuss potential physiological relevance of the egress pathway. Could this actually benefit the parasite in vivo by escaping the toxic intracellular environment of an IFNγ-primed cells? Would ejected parasites be more easily killed by phagocytes?

We added a section in the Discussion covering the potential physiological relevance of RARRES3.

Suggested experiments/ data analysis– REF [57] showed timelapse microscopy capturing parasite egress in IFNγ primed cells. Similar experiments monitoring egress in real time in controls vs RARRES3 overexpression and/or WT vs RARRES3 KO cells would provide direct evidence in support of the proposed mechanistic model and therefore significantly strengthen the paper. Current evidence is entirely circumstantial.

As indicated above, we conducted a live imaging experiment to directly observed parasite egress in RARRES3 and FLUC ectopically expressing A549s. The data is presented in figure 6C and videos 1-2.

– Host-driven/ host-protective egress pathways have been reported for other infectious agents (cryptococcus ['vomocytosis'], UEPC,.…) – it would broaden the impact of the study, if the authors were to test whether RARRES3 is also involved in the egress of one or two of these pathogens. Regardless of the outcome of these studies, the results would be highly informative. Authors have done these types of comparisons in previous follow-up studies to similar overexpression screens (see Listeria ISG overexpression screen and follow-up with Shigella).

We agree that it would be interesting to test other infectious agents. However, we believe that this is beyond the scope of the current paper and should be addressed in a future study.

– Provide statistical comparison of complemented KOs vs controls (KO + FLUC) under IFNγ primed conditions (Figure 5B). If study is underpowered, conduct more repeats. If no complementation is achieved (is achievable), then explain why.

The complemented KO is not significantly different from WT demonstrating complementation. A “ns” comparison between these bars has been added to the figure for clarity.